# USP5-Beclin 1 axis overrides p53-dependent senescence and drives *Kras*-induced tumorigenicity

Juan Li [1,5], Yang Wang [1,5] ✉, Yue Luo[1], Yang Liu[1], Yong Yi [1], Jinsong Li[1], Yang Pan [1], Weiyuxin Li[1], Wanbang You[1], Qingyong Hu[1], Zhiqiang Zhao[1], Yujun Zhang[1], Yang Cao [1], Lingqiang Zhang [2], Junying Yuan [3] & Zhi-Xiong Jim Xiao [1,4] ✉

Non-small cell lung cancers (NSCLC) frequently contain *KRAS* mutation but retain wild-type *TP53*. Abundant senescent cells are observed in premalignant but not in malignant tumors derived from the *Kras*-driven mouse model, suggesting that *KRAS* oncogenic signaling would have to overcome the intrinsic senescence burden for cancer progression. Here, we show that the nuclear Beclin 1-mediated inhibition of p53-dependent senescence drives *Kras*-mediated tumorigenesis. *KRAS* activates USP5 to stabilize nuclear Beclin 1, leading to MDM2-mediated p53 protein instability. *Kras^G12D* mice lacking Beclin 1 display retarded lung tumor growth. Knockdown of *USP5* or knockout of *Becn1* leads to increased senescence and reduced autophagy. Mechanistically, *KRAS* elevates ROS to induce USP5 homodimer formation by forming the C195 disulfide bond, resulting in stabilization and activation of USP5. Together, these results demonstrate that activation of the USP5-Beclin 1 axis is pivotal in overriding intrinsic p53-dependent senescence in *Kras*-driven lung cancer development.

Oncogenic *RAS* is a powerful driving force for tumorigenesis through promoting cell proliferation, differentiation, and survival[1]. However, activation of oncogenic signals, including oncogenic *RAS*, *BRAF*, *MYC*, or *ERBB2*, frequently induces cellular senescence, named oncogene-induced senescence (OIS), which is usually p53-dependent[2]. Abundant evidence indicates that OIS can serve as a barrier in blocking uncontrolled proliferation and that the OIS-imposed barrier has to be overridden for the tumorigenesis to proceed[3]. For instance, senescent cells exist in various human primary tumors, including colon adenomas, astrocytoma, neurofibroma, as well as early-stage prostate tumors[3]. In a *Kras^G12V* mice model, senescent cells are readily detected in premalignant lung adenomas but rarely in malignant ductal adenocarcinomas[4].

Autophagy is a conserved self-degradative pathway essential for protein homeostasis, survival, and development. Dysregulation of autophagic pathways has been connected to a variety of human diseases, including cancer and neurodegeneration[5]. Abundant evidence indicates that autophagy plays an important role in activated Ras-induced tumorigenesis[6]. The *Kras^G12V*-driven salivary duct carcinoma progression is severely hindered by a deficiency of the essential autophagy gene *Atg5*[7]. Similarly, *Kras^G12D*-driven NSCLC is inhibited by *Atg7* deletion[8]. Notably, in cultured cell systems, *KRAS^G12V* induces expression of *ATG5* and *ATG7* in immortalized human epithelial MCF-10A cells in promoting cell proliferation and tumor growth, which are effectively rescued by eliminating reactive

[1]Center of Growth, Metabolism and Aging, College of Life Sciences, Sichuan University, Chengdu 610065, China. [2]State Key Laboratory of Proteomics, National Center for Protein Sciences (Beijing), Beijing Institute of Lifeomics, Beijing 100850, China. [3]Interdisciplinary Research Center on Biology and Chemistry, Shanghai Institute of Organic Chemistry, Chinese Academy of Sciences, 100 Haike Rd, Pudong, Shanghai 201210, China. [4]State Key Laboratory of Biotherapy, Sichuan University, Chengdu 610041, China. [5]These authors contributed equally: Juan Li, Yang Wang. ✉e-mail: wangy90@scu.edu.cn; jimzx@scu.edu.cn

oxygen species (ROS) with NAC or ectopic expression of *CAT* encoding catalase[9].

Beclin 1 is a central player in autophagy and constitutes a molecular platform for the regulation of autophagosome formation and maturation. Posttranslational modification (PTM) is important for the regulation of Beclin 1 function, including phosphorylation, acetylation, and ubiquitination[10,11]. Akt can interact with and phosphorylate S234 and S295 of Beclin 1 to inhibit autophagy[10]. AMPK promotes S90 and S93 phosphorylation of Beclin 1 to induce autophagy in response to glucose starvation[10]. Beclin 1 can be acetylated by p300 and deacetylated by SIRT1 at K430 and K437, resulting in the modulation of autophagosome maturation[11]. Ubiquitination regulates Beclin 1 in a complex manner. The K6-, K27- and K63-linked ubiquitination of Beclin 1 promotes Beclin 1 protein stability, whereas K48-linked ubiquitination of Beclin 1 leads to Beclin 1 degradation and inhibition of autophagy. In addition, the ubiquitination of Beclin 1 can impact its activity. TRAF6-mediated K63-ubiquitination of Beclin 1 does not affect Beclin 1 levels but inhibits its interaction with Bcl-2 to promote autophagy, which can be reversed by A20 or USP14[10].

Protein stability is mainly controlled by the ubiquitin-proteasome system, which is regulated by a cascade of enzymatic reactions, including Ub-activating (E1), Ub-conjugating (E2), Ub-ligating (E3) enzymes, and deubiquitinating enzymes (DUBs). USPs (ubiquitin-specific proteases), a subclass of DUB, play important roles in the regulation of a variety of biological and pathophysiological processes[12]. It has been shown that USP5 is often upregulated in NSCLC[13] and in other human cancers[14]. USP5 can facilitate cancer progression, including NSCLC, through deubiquitination and stabilization of several downstream target proteins, including β-catenin, SLUG, FoxM1, Cyclin D1, and PD-L1[13,15–19]. However, it remains unclear how USP5 is regulated and the role of USP5 in *KRAS*-induced tumorigenesis.

Reactive oxygen species (ROS) consist of a variety of oxidant molecules, including superoxide ($O_2^{•-}$), hydroxyl radical (•OH), nitric oxide (NO•), hydrogen peroxide ($H_2O_2$), singlet oxygen($^1O_2$), and ozone/trioxygen ($O_3$). These molecules possess vastly different properties and perform diverse biological functions ranging from signaling to causing cell death. It is well established that $H_2O_2$ is a major signaling molecule that leads to protein modification and modulation of signaling events. $H_2O_2$ can reversibly oxidize cysteine residues in proteins, including intramolecular or intermolecular disulfide formation and other modifications such as glutathionylation or persulfate of the reactive cysteine, thereby controlling their stability and activity[20].

Abundant evidence indicates that ROS is intimately involved in tumorigenesis in a dose and context-dependent manner. A moderate increase in ROS can promote cell proliferation and cell survival, whereas excessive amounts of ROS lead to cell damage and death[21]. It is well known that many cancers accumulate genetic alterations that promote the production of ROS, resulting in elevated ROS levels in cancer cells. Activation of oncogenes, such as *KRAS*, *BRAF*, and *MYC*, often leads to increased ROS generation, while it can also drive compensatory antioxidant responses[22], that can be mediated by increasing cystine transport and by inducing the expression of NRF2, a critical transcriptional factor that can induce the expression of many antioxidant genes. NRF2 is essential for tumorigenesis in *Kras*-driven mouse lung and pancreatic cancer models[21]. The elevated ROS production and the maintenance of redox homeostasis are essential for *Kras*-driven cell growth and tumorigenesis[21]. Yet, the direct ROS-targeted protein(s) that are critical for *Kras*-driven tumorigenesis remain unknown.

Here, we show that *KRAS* upregulates USP5 protein stability and enzymatic activity through ROS-mediated protein dimerization. Activated USP5 stabilizes Beclin 1, which in turn facilitates autophagy and promotes p53 protein instability. Inhibition of either USP5 or Beclin 1 suppresses autophagy and imposes a p53-dependent senescence burden resulting in blockage of *Kras*-mediated NSCLC growth.

## Results

### Ablation of pro-autophagy *USP5* downregulates Beclin 1 to upregulate p53 and cellular senescence in the suppression of lung tumor growth

Beclin 1 is a core protein in the initiation of autophagosome formation, a key event in the autophagy pathway[23]. Mounting evidence indicates that autophagy is critically important in promoting tumorigenesis, including activated KRas-driven lung tumor growth[24]. The Beclin 1 protein stability is modulated by ubiquitin E3 ligases and DUBs[10]. Thus, we screened DUBs for Beclin 1 that may have clear clinical relevance to lung cancer patients. We constructed stable 293FT-GFP-RFP-LC3 reporter cells, which displayed uniform and evenly distributed red/ green fluorescence (Supplementary Fig. S1a). A panel of plasmids expressing Flag-tagged USPs from a DUB library was transiently expressed in 293FT-GFP-RFP-LC3 reporter cells, followed by FACS analyses to score for intensity changes in GFP fluorescence, a marker reflecting levels of autophagy. As shown in Supplementary Fig. S1b and Table S3, FACS analyses showed that ectopic expression of a subset of USP genes significantly impacted GFP fluorescence intensity, including previously identified autophagy-modifying USPs (USP1, USP8, USP12, USP19, USP20, USP22, and USP36)[25–31]. Notably, several new USPs were able to significantly ($P < 0.001$) increase autophagy as indicated by reduced GFP fluorescence, including USP1, USP5, USP7, USP12, USP18, USP20, and USP46, among which, only high USP5 expression was associated with poor OS (Supplementary Fig. S1c and Table S3). These results suggest that USP5 is a pivotal gene promoting autophagy with a link to poor clinical outcomes in lung cancer patients (Supplementary Fig. S1d).

To verify the pro-autophagic function of USP5, we examined the effects of USP5 or USP5[W209A] [32], a mutant defective in deubiquitinating function, on autophagosome formation (LC3 puncta) in A549-RFP-GFP-LC3 or H292-RFP-GFP-LC3 reporter cells. RFP-GFP-LC3 was used to monitor the autophagic flux in cells[33]. RFP-GFP-LC3 showed GFP as well as RFP signals in autophagosomes which are before fusion with lysosomes. The presence of only RFP signal reflects the subsequent formation of autolysosomes[33]. As shown in Fig. 1a, wild-type USP5, but not USP5[W209A], significantly increased both autophagosome (GFP-RFP-LC3 puncta) and autolysosomes (RFP-LC3 puncta), concomitant with increased expression of Beclin 1 and LC3-II, and decreased the levels of p62/SQSTM1 protein (Fig. 1b). Furthermore, knockdown of *USP5* led to significantly reduced expression of Beclin 1 and LC3-II, and increased expression of p62/SQSTM1. Notably, silencing of *USP5* dramatically upregulated levels of p53 and p21, a p53 downstream cell cycle inhibitor, concomitant with induced cellular senescence and inhibited colony formation of A549 and H292 cells (Fig. 1c–e and Supplementary Fig. S2a). In addition, the knockdown of *USP5* did not induce apoptosis in A549 and H292 cells (Supplementary Fig. S2b, c). Similar observations, including cellular senescence and inhibited colony formation, were obtained using a selective USP5 inhibitor, WP1130[34] (Supplementary Fig. S2d–f). Moreover, WP1130-induced cellular senescence, inhibited autophagy, and reduced cell growth were fully recused by ectopic expression of WT USP5, but not USP5[W209A] (Supplementary Fig. S2g–i).

To investigate whether Beclin 1 plays a role in cellular senescence induced by silencing of *USP5*, a set of rescuing experiments was performed. As shown in Fig. 1f–h, ectopic expression of Beclin 1, but not Beclin 1[S90+93A], a mutant defective in the initiation of autophagy[35], significantly reversed the expression of p53 and p21 upregulated by knockdown of *USP5*, accompanied with significantly reduced cellular senescence and recovered ability of colony formation.

We further assessed the role of the USP5-Beclin 1 in NSCLC growth in the xenograft mouse model. A549-Antares2 cells[36] bearing *USP5* shRNA were stably expressed Beclin 1 or Beclin 1[S90+93A], which were transplanted subcutaneously into flanks of nude mice. As shown in Fig. 1i–n and Supplementary Fig. S2j, depletion of *USP5* led to

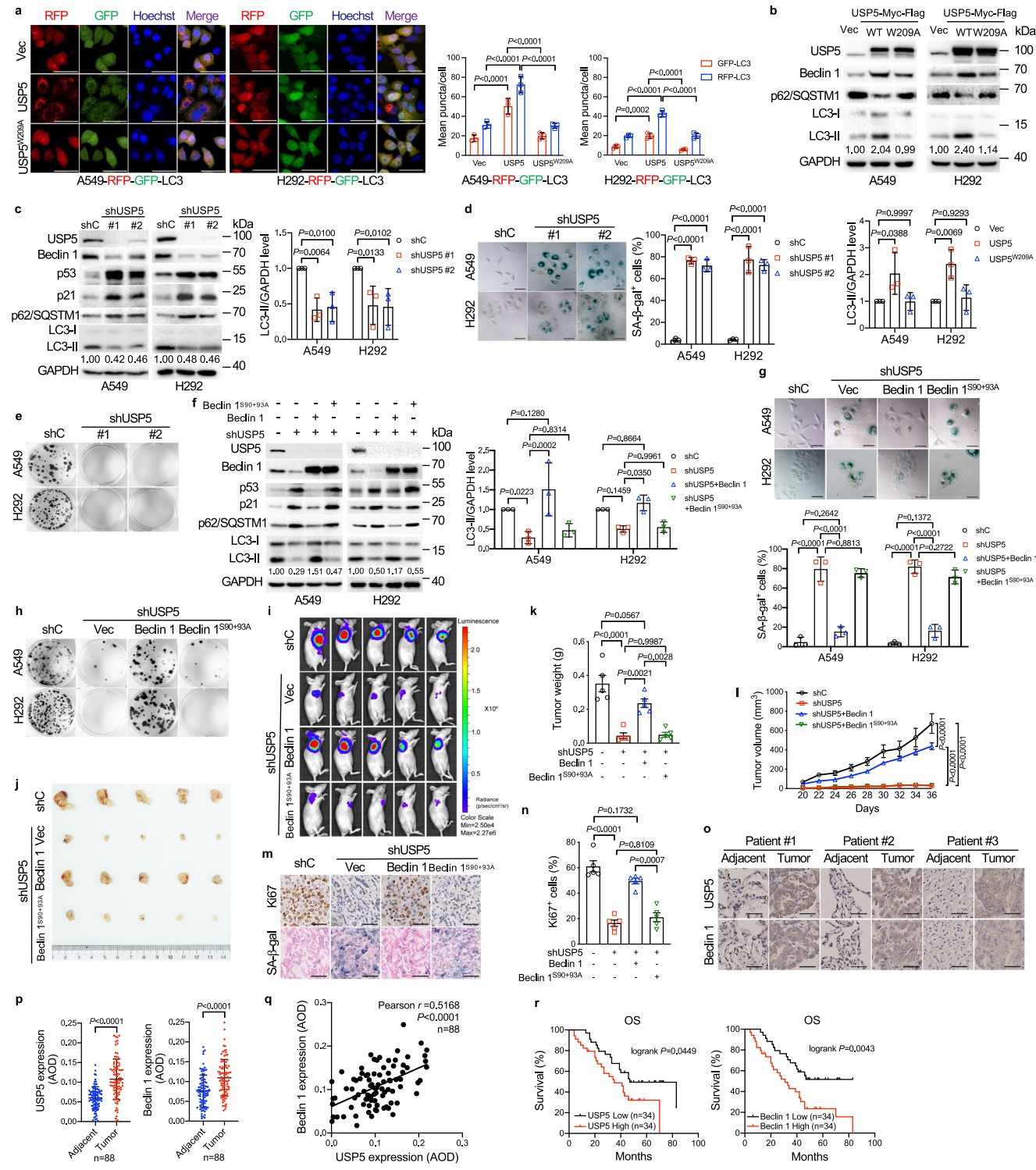

significantly reduced xenograft tumor growth, concomitant with elevated p53, p62/SQSTM1, and SA-β-gal staining, and markedly reduced Ki67. Notably, depletion of *USP5* also led to reduced autophagy, as evidenced by increased expression of p62/SQSTM1 and reduced expression of Beclin 1, and LC3 aggregates. Importantly, ectopic expression of Beclin 1, but not Beclin 1$^{S90+93A}$, remarkably rescued the tumor growth inhibited by *USP5* depletion. Collectively, these results demonstrate that depletion of *USP5* suppresses lung tumor growth via reducing Beclin 1 expression and activation of p53, attributable to reduced autophagy and increased senescent cell burden.

Abundant evidence indicates that USP5 expression is elevated in NSCLC cancers[13,17–19], while the correlation of Beclin 1 protein expression with NSCLC remains debatable[37]. We thus examined the expression of USP5 and Beclin 1 in human lung adenocarcinoma biopsy samples and paired adjacent tissue via IHC, followed by analyses of Average Optical Density (AOD)[38]. As shown in Fig. 1o, p, the expression of USP5 or Beclin 1 was evidently elevated as compared with the matched adjacent tissues. Statistical analyses revealed a positive correlation between USP5 and Beclin 1 (Fig. 1q). Moreover, lung cancer patients with either high USP5 or Beclin 1 protein levels had a worse OS

**Fig. 1 | USP5 modulates Beclin 1 protein expression to regulate autophagy, cancer cell senescence, and lung tumor growth. a, b** Stable A549-RFP-GFP-LC3 or H292-RFP-GFP-LC3 reporter cells expressing either WT USP5 or USP5^W209A were analyzed for LC3 puncta formation under a confocal microscope. Quantification of RFP-LC3 and GFP-LC3 puncta was shown (**a**). A549 and H292 cells stably expressing WT USP5 or USP5^W209A were subjected to western blot (**b**). **c–h** A549 and H292 cells stably expressing shRNA specific for *USP5* (#1 or #2) or control shRNA (shC) (**c–e**) or A549 and H292 cells stably expressing shUSP5 (shUSP5 #1), shUSP5 + WT Beclin 1 or Beclin 1^S90+93A (**f–h**) were subjected to western blot, SA-β-gal staining or colony formation assays. **i–n** For tumor growth assay, A549-Antares2 cells (2 × 10⁶) stably expressing shUSP5, shUSP5+Beclin 1 or shUSP5 + Beclin 1^S90+93A were subcutaneously transplanted into nude mice (*n* = 5/group). Diphenylterazine (DTZ) was injected into tumor regions in anesthetized mice and bioluminescence was subsequently imaged (**i**), and the harvested tumors were photographed (**j**), and

weighted (**k**). The tumor volumes of cell-derived xenografts were quantified every other day (**l**). The tumors were subjected to immunohistochemical (IHC) staining for Ki67 and SA-β-gal (**m**), and the percentages of Ki67-positive cells were presented (**n**). **o–r** Human lung adenocarcinoma tissue microarrays consisting of lung tumor and adjacent normal tissues (*n* = 88 pairs) were subjected to IHC for USP5 and Beclin 1 (**o**), with quantitative analyses using average optical density (AOD) (**p**). The two-tailed Pearson correlation between USP5 with Beclin 1 protein levels (**q**) and the correlation of USP5 or Beclin 1 protein levels with Overall Survival in lung cancer patients (**r**) were analyzed. Experiments were performed three times independently (**a–h**). Data were presented as mean ± SD (**a–d, f, g, p**) or SEM (**k, l, n**). Comparisons were performed with two-way (**a–d, f, g, k, l, n**) ANOVA with Tukey's test analysis for multiple comparisons and two-tailed Student's *t*-test (**p**). Scale bar = 50 μm.

(Fig. 1r). Together, these findings suggest that the USP5-Beclin 1 pathway plays a critical role in lung adenocarcinoma development.

## USP5 binds to and promotes stabilization of Beclin 1

To explore the mechanism by which USP5 upregulates Beclin 1 expression, we first performed qPCR analyses. As shown in Fig. 2a, ectopic expression of either USP5 or USP5^W209A had little effect on the steady-state *BECN1* mRNA levels. By contrast, ectopic expression of USP5, but not USP5^W209A, significantly extended the protein half-life of Beclin 1 (Fig. 2b). Consistently, depletion of *USP5* significantly shortened the half-life of Beclin 1 protein (Fig. 2c). We next examined the interaction between USP5 and Beclin 1. As shown in Fig. 2d, the endogenous USP5-Beclin 1 protein complex was readily detected. Beclin 1 (Δ270−450), lacking the ECD-C domain, was unable to interact with USP5, suggesting that the ECD-C domain of Beclin 1 is the binding site for USP5 (Fig. 2e). Furthermore, USP5 and Beclin 1 were co-localized in both nuclei and cytoplasm (Fig. 2f).

We then investigated the impact of USP5 on the deubiquitination of Beclin 1. As shown in Fig. 2g, ectopic expression of USP5, but not USP5^W209A, effectively removed the ubiquitin chains of Beclin 1. Conversely, silencing of *USP5* significantly facilitated Beclin 1 polyubiquitination (Fig. 2h). Furthermore, USP5 could effectively remove the K48-linked polyubiquitination, but not K63-linked polyubiquitination (Fig. 2i). Together, these results indicate that USP5, acting as a ubiquitin-specific protease, binds to and deubiquitinates K48-linked polyubiquitination of Beclin 1, resulting in the stabilization of Beclin 1.

## Depletion of *USP5* leads to stabilization of p53 protein, resulting in cellular senescence and suppression of tumor growth

Our aforementioned data indicate that ablation of *USP5* leads to destabilization of Beclin 1 protein, concomitant with increased p53 protein expression and cellular senescence. To investigate the underlying mechanism, we examined the effects of USP5 on the steady-state p53 mRNA or protein levels. As shown in Supplementary Fig. S3a, the knockdown of *USP5* had little effect on the *p53* mRNA level. However, silencing of *USP5* significantly prolonged the p53 protein half-life (Fig. 3a), indicating that USP5 regulates p53 protein stability.

We next investigated whether p53 is required for *USP5* knockdown-induced cellular senescence. As shown in Fig. 3b–d, ablation of *p53* completely blocked *USP5* knockdown-induced upregulation of both p21 and SA-β-gal activity and inhibition of colony formation. In addition, ablation of *USP5* could not induce senescence in lung cancer cells harboring a mutant *p53* allele or *p53* null (Supplementary Fig. S3b). To further investigate the role of the USP5-p53 axis in NSCLC growth in the xenograft mouse model, A549-Antares2 cells, bearing *USP5* shRNA and stably expressing *p53* shRNA, were transplanted subcutaneously into flanks of nude mice. As shown in Fig. 3e–j and Supplementary Fig. S3c, while *USP5* depletion drastically inhibited tumor growth, simultaneous ablation of *p53* remarkably restored tumor growth, concomitant with increased Ki67 and autophagosomes

as well as decreased p62/SQSTM1. Importantly, ablation of *p53* significantly reduced SA-β-gal activity in the tumors. These results strongly suggest that USP5-mediated tumor growth inhibition is dependent on wild-type p53.

## Nuclear Beclin 1 enhances MDM2-p53 interaction to promote p53 degradation resulting in suppression of *USP5* knockdown-induced cellular senescence

To further investigate the role of Beclin 1 in cellular senescence, we employed specific shRNAs to effectively ablate *BECN1*. As shown in Fig. 4a–c, ablation of *BECN1* significantly upregulated the expression of p53 and p21, concomitant with the induction of cellular senescence and inhibition of colony formation. Silencing of *p53* completely blocked Beclin 1 ablation-induced cellular senescence and cell growth inhibition (Fig. 4d–f). Additionally, similar to the ablation of *USP5*, the ablation of *BECN1* could not induce senescence in lung cancer cells harboring a mutant *p53* allele or *p53* null (Supplementary Fig. S4a). Notably, p53 was co-localized with Beclin 1 in nuclei (Fig. 4g and Supplementary Fig. S4b). Silencing of either *BECN1* or *USP5* significantly increased nuclear p53 expression (Fig. 4h, i).

Furthermore, WT Beclin 1, but not Beclin 1^L184+187A, defective in nuclear export and thereby unable to promote autophagy[39], could promote autophagy, as expected (Fig. 4j and Supplementary Fig. S4c). However, both Beclin 1 and Beclin 1^L184+187A were able to rescue the expression levels of p53 and p21 induced by *USP5* ablation, resulting in suppression of *USP5* ablation-induced cellular senescence and cell growth inhibition (Fig. 4j–l). These results indicate that cytoplasmic Beclin 1 regulates autophagy while nuclear Beclin 1 regulates cellular senescence and cell growth via modulation of the p53-p21 pathway and that both autophagy and senescence pathways are significantly impacted by USP5.

We further investigated the molecular basis with which Beclin 1 regulates p53 expression and cellular senescence. As shown in Supplementary Fig. S4d, e, while the knockdown of *BECN1* did not significantly affect steady-state *p53* mRNA levels, it dramatically prolonged the p53 protein half-life. Since MDM2 is the most critical ubiquitin E3 ligase targeting p53 for proteasomal degradation, we then examined whether Beclin 1 regulates p53 protein stability via modulation of MDM2. The stable trimeric protein complexes consisting of Beclin 1, p53, and MDM2 were readily detectable in vivo (Fig. 4m) and in vitro (Fig. 4n, o). In addition, the ablation of *BECN1* significantly reduced MDM2 interaction with p53 (Fig. 4p) but did not affect the expression of MDM2 (Supplementary Fig. S4f). Importantly, Beclin 1 facilitated MDM2 interaction with p53 in a dose-dependent manner in vitro (Fig. 4q) and in vivo (Fig. 4t). Furthermore, co-IP assays showed that the CCD domain of Beclin 1 interacted with the segment (AA 154-221) of MDM2 (Fig. 4r, s). Notably, the BH3 domain of Beclin 1 is bound to p53 (Supplementary Fig. S4g), in keeping with a previous report[40]. Ectopic expression of WT Beclin 1 significantly enhanced the interaction between MDM2 and p53, whereas either ΔBH3 defective in p53-

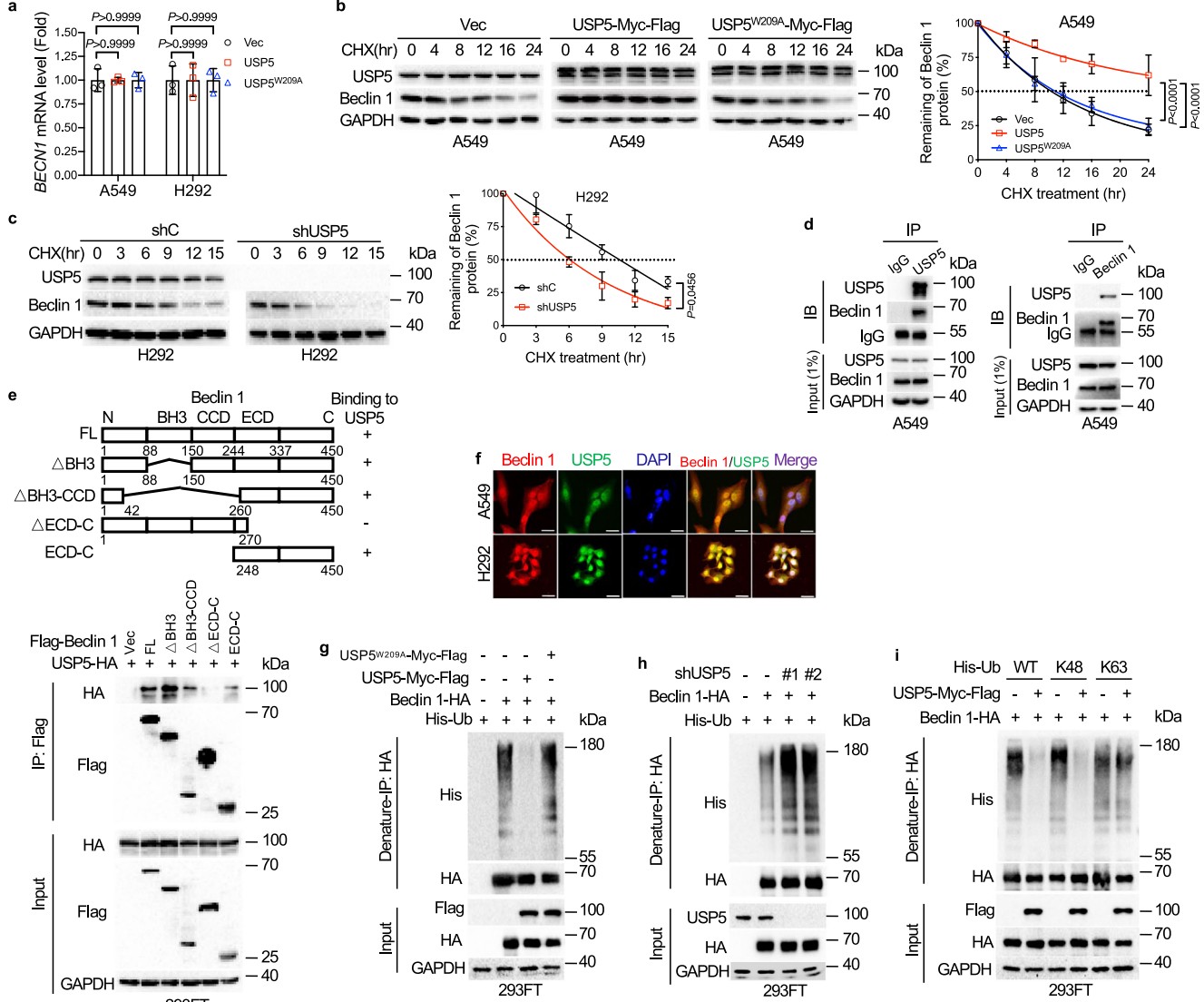

**Fig. 2 | USP5 deubiquitinates and stabilizes Beclin 1. a** A549 or H292 cells stably expressing USP5 or USP5[W209A] were subjected to qPCR analyses for *BECN1* mRNA levels. **b, c** A549 cells stably expressing USP5 or USP5[W209A] were treated with 50 µg/mL cycloheximide (CHX) for the indicated time intervals (**b**). H292 cells stably expressing shUSP5 were treated with 50 µg/mL CHX for the indicated time intervals (**c**). The protein half-life plots were obtained by quantifying relative Beclin 1/GAPDH protein bands derived from the corresponding western blots. **d** A549 cells were subjected to co-immunoprecipitation (co-IP) using either a specific antibody for USP5, Beclin 1 or IgG control followed by western blot analyses. **e** A schematic diagram of Beclin 1-USP5 interaction (upper panel). 293FT cells were transfected with USP5-HA and Flag-Beclin 1 (FL) or an indicated mutant construct. Total cell lysates were subjected to IP-western blot analyses. **f** A549 and H292 cells were subjected to immunofluorescent (IF) staining for endogenous Beclin 1 (red) or USP5 (green) and counter-staining nuclei with DAPI (blue). Scale bar = 50 µm. **g–i** To examine the effects of USP5 on ubiquitination of Beclin 1, 293FT cells were co-transfected with indicated expressing plasmids. Cells were treated with 20 µM MG132 for 6 h before collection. Ubiquitination of Beclin 1 was examined by denature-IP-western analyses. Experiments were performed three times independently (**a–i**). Data were presented as mean ± SD (**a–c**). Comparisons were performed with two-way ANOVA with Tukey's (**a**, **b**) or Bonferroni's (**c**) test.

interaction or ΔCCD defective in MDM2 interaction failed to do so (Fig. 4t). Moreover, similar to WT Beclin 1, Beclin 1[L184+187A] exclusively localized in the nuclei formed stable protein complexes with p53. By sharp contrast, Beclin 1[S90+93A], a mutant protein exclusively localized in the cytosol and defective in the initiation of autophagy[35], showed little interaction with p53 (Supplementary Fig. S4h), consistent with our observations that, unlike WT Beclin 1, Beclin 1[S90+93A] was unable to rescue either the upregulation of p53 or the induction of cellular senescence induced by knockdown of *USP5* (Fig. 1f). Additionally, ablation of *USP5* or *BECN1* significantly decreased the ubiquitination of p53, which could be rescued by ectopic expression of MDM2 (Supplementary Fig. S4i, j). Together, these results suggest that Beclin 1 acts as an adapter to promote the trimeric complex formation, thereby enhancing MDM2-mediated p53 degradation (Fig. 4u).

## ROS mediates *KRas*[G12V]-induced dimerization, stabilization, and activation of USP5

Our aforementioned results indicate that ablation of *USP5* leads to inhibition of xenograft lung tumor growth (Fig. 1i–n), suggesting that USP5 functions as an oncogene to promote lung cancer development. Since activation of KRas is a driving force in the initiation and development of NSCLC, we hypothesized that USP5, which regulates both autophagy and senescence, plays an important role in *KRAS*-driven NSCLC development. Thus, we first examined the effects of *KRAS*[G12V] on the expression and enzymatic activity of USP5. In H292 cells (wild-type *KRAS*) stably expressing doxycycline-inducible *KRAS*[G12V] (TRE:*KRAS*[G12V]), induction of *KRAS*[G12V] expression led to elevated ROS production at 12 h (Supplementary Fig. S5a), and upregulated USP5 deubiquitinating enzymatic activity as evidence by the DUB-labeling assay[34], followed by

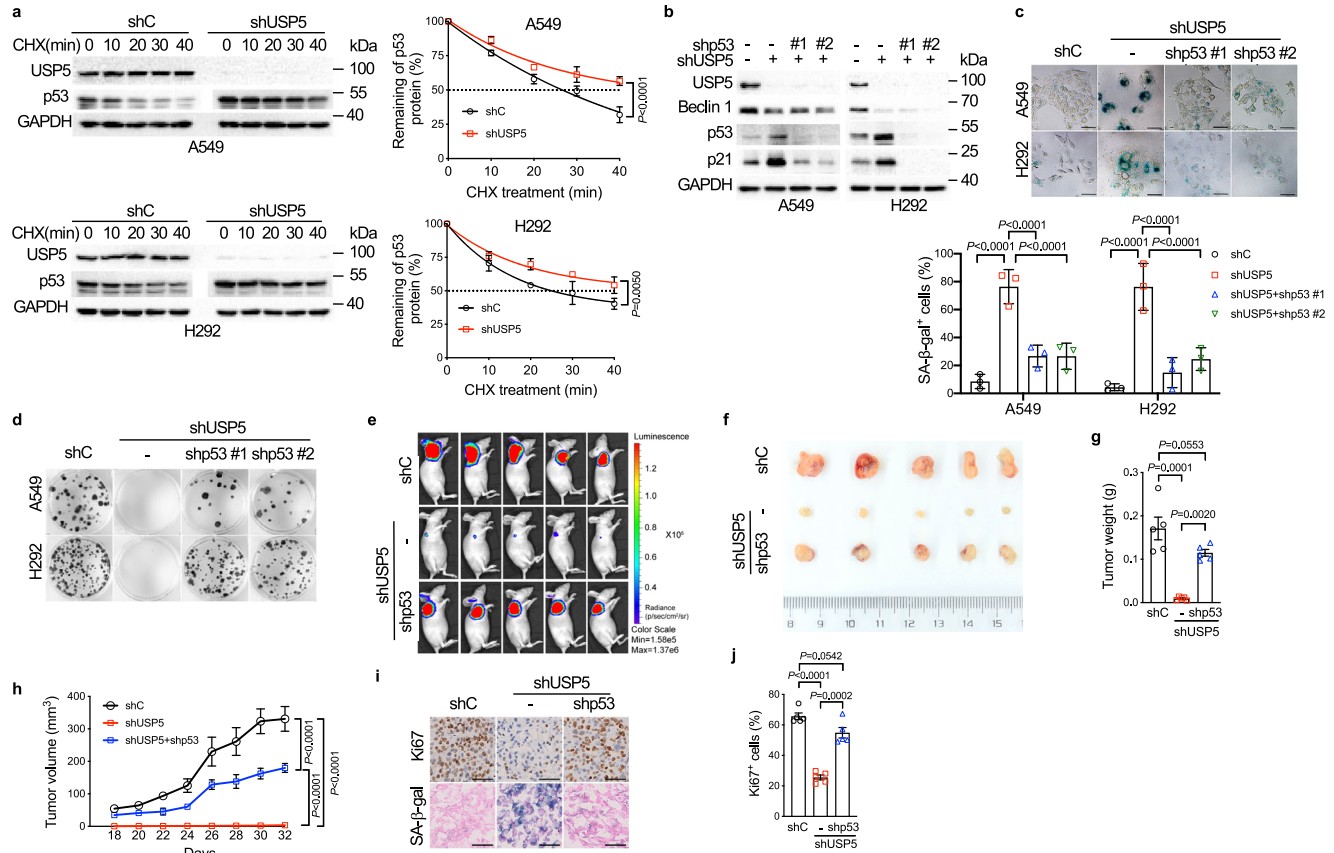

**Fig. 3 | Ablation of *USP5* leads to stabilization of p53 resulting in senescence and suppression of tumor growth in vivo. a** A549 or H292 cells stably expressing shUSP5 or control shRNA (shC) were treated with 50 µg/mL CHX for the indicated time intervals before collection. The protein half-life plots were shown. **b–d** A549 or H292 cells stably expressing shUSP5 were infected with lentiviral-shp53 (#1 or #2) or control shRNA (shC) followed by western blot analyses (**b**), SA-β-gal staining (**c**), or colony formation assays (**d**). Scale bar = 50 µm. **e–j** A549-Antares2 cells (2 × 10⁶) stably expressing shUSP5 and shUSP5 + shp53 were subcutaneously transplanted

into nude mice (*n* = 5/group). DTZ was injected into tumor regions in anesthetized mice and bioluminescence was subsequently imaged using Caliper IVIS Lumina III (**e**). Tumors were excised, photos taken (**f**), and weighted (**g**). Tumor sizes were measured every other day (**h**). IHC analyses were performed for Ki67 and SA-β-gal (**i, j**). Experiments were performed three times independently (**a–d**). Data were presented as mean ± SD (**a, c**) or SEM (**g, h, j**). Comparisons were performed with two-way ANOVA with Bonferroni's (**a**) or Tukey's (**c, g, h, j**) test. Scale bar = 50 µm.

a detectable increase in protein expression of Ras and USP5 protein levels at 48 h, accompanied by altered expression of Beclin 1, p62/SQSTM1, LC3-II, and p53 (Fig. 5a).

We then examined the impact of ROS inducer, either $H_2O_2$ or piperlongumine (PL)[41], on the USP5 expression and activity. As shown in Fig. 5b, either $H_2O_2$ or PL significantly elevated USP5 protein expression and enzymatic activity, concomitant with dramatic upregulation of Beclin 1 expression. We further investigated the role of ROS in *KRAS^{G12V}*-induced upregulation of USP5. Again, *KRAS^{G12V}* elevated ROS levels (Supplementary Fig. S5b), whereas elimination of ROS by NAC (*N*-acetylcysteine) effectively blocked *KRAS^{G12V}*-mediated upregulation of USP5 enzymatic activity and protein levels of USP5 and Beclin 1 (Fig. 5c). Notably, *KRAS^{G12V}*-mediated USP5 protein stabilization was completely reversed by NAC treatment (Fig. 5d), indicating that ROS is critically important in the *KRAS^{G12V}*-mediated modulation of USP5. Notably, the elimination of ROS by NAC could also reduce basal levels of USP5 protein expression and enzymatic activity in H292 cells lacking activated KRas (Supplementary Fig. S5c), indicating that ROS is critically important in USP5 protein expression and enzymatic activity.

We next investigated the molecular basis by which *KRAS^{G12V}*-induced ROS affects USP5 protein stability. It has been well documented that disulfide-based dimerization is important for protein stability and ROS can promote disulfide bond formation[42]. We then examined the possible USP5 dimer formation by observing the USP5 protein migratory shift on SDS-PAGE in the presence or absence of

dithiothreitol (DTT), a disulfide-reducing agent. As shown in Fig. 5e and Supplementary Fig. S5d, *KRAS^{G12V}* induced the appearance of slower mobility of USP5 protein on SDS-PAGE in the absence of DTT, which was completely rescued by NAC. Furthermore, $H_2O_2$ could induce the slower mobility of USP5 (Fig. 5f) and promote recombinant USP5 protein dimerization in vitro, in a dose-dependent manner (Supplementary Fig. S5e and Fig. 5g). Together, these results suggest that ROS promotes USP5 protein dimerization resulting in elevated protein stabilization and enzymatic activity.

To explore the structural basis of USP5 protein for dimerization, we analyzed crystal structures of USP5 aimed to identify sulfhydryl groups exposed to the surface accessible for disulfide formation. As shown in Supplementary Fig. S5f, g, based on the partial 3D structure of USP5 derived from X-ray diffraction (Protein Data Bank entry 3IHP), analyses using *PyMOL*[43] projected seven cysteines, including C46, C195, C202, C335, C703, C793, and C815, being located on the surface of USP5. Notably, a recent proteomics analysis identified that only C195 is engaged in the formation of disulfide bonds[44]. Furthermore, we predicted full-length USP5 protein structures by *Alphafold2*[45] and projected the USP5 dimerization by the *Gromacs*[46], in which a disulfide bond is formed between C195 residues of two monomers (Supplementary Fig. S5h).

To verify the role of C195 in ROS-induced USP5 dimerization, we examined the effects of *KRAS^{G12V}* or $H_2O_2$-induced USP5 dimerization. As shown in Fig. 5h, i, *KRAS^{G12V}* or $H_2O_2$ could induce mobility shift of WT

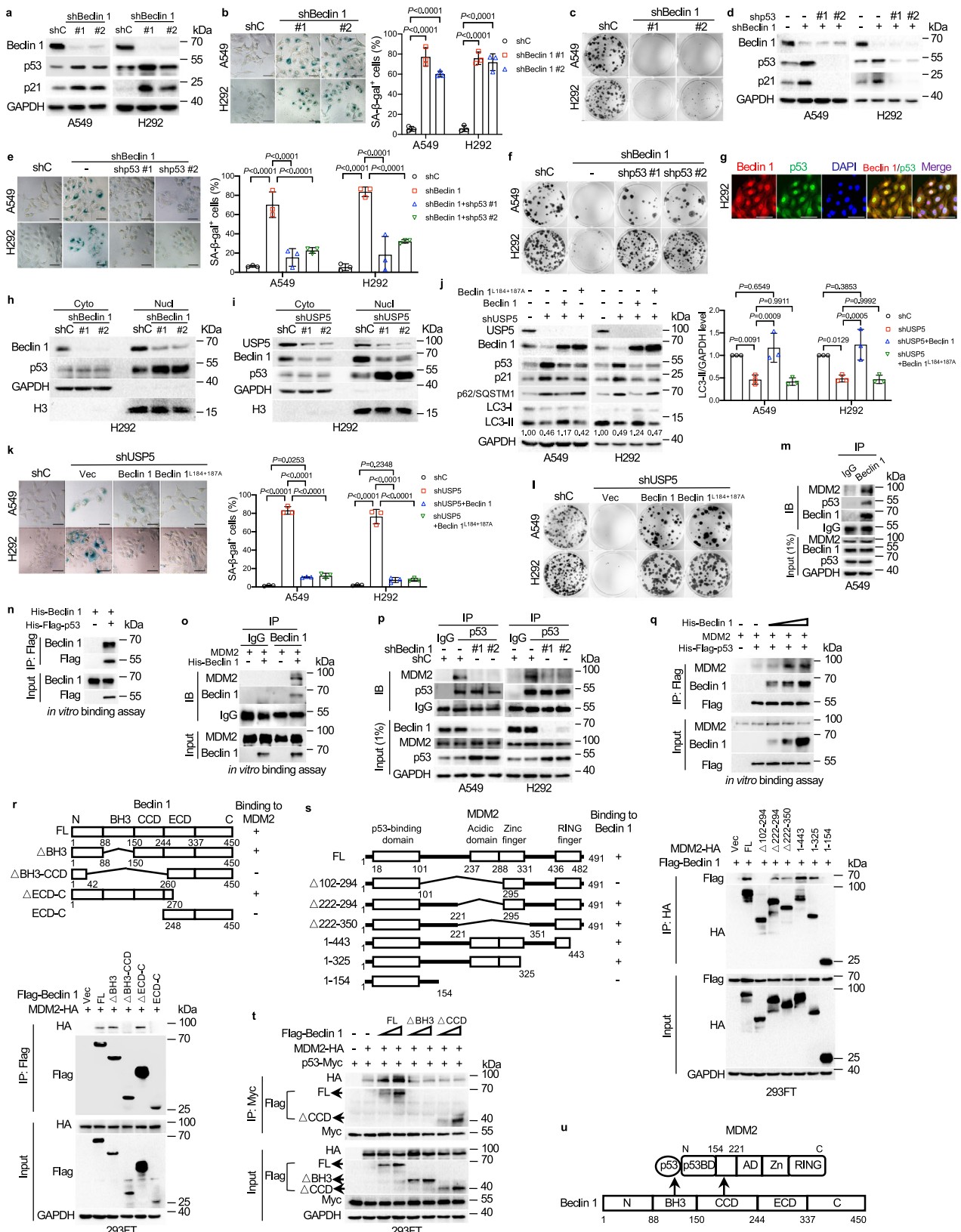

USP5, but not USP5^C195A mutant protein. Furthermore, the purified recombinant USP5 treated with $H_2O_2$, was analyzed by analytical FPLC gel filtration, which showed the formation of dimerization, whereas there was no dimerization of USP5^C195A samples detected under the same setting (Fig. 5j). Consistently, western blot analyses showed that the $H_2O_2$ treatment of purified recombinant WT USP5, but not USP5^C195A

proteins, could readily form dimers in vitro (Fig. 5k). These results indicate that KRas^G12V induces USP5 dimerization through ROS-mediated formation of the C195 disulfide bond between two monomers.

We then investigated the biological significance of KRas^G12V-induced USP5 dimerization. WT USP5, but not USP5^C195A, effectively removed ubiquitin chains of Beclin 1 in vivo (Supplementary Fig. S5i)

**Fig. 4 | Nuclear Beclin 1 enhances MDM2-p53 interaction to destabilize p53 and inhibits the USP5 knockdown-mediated senescence. a–c** A549 or H292 cells stably expressing shBeclin 1 (#1 or #2) were subjected to western blot (**a**), SA-β-gal staining (**b**), or colony formation assays (**c**). **d–f** A549 or H292 cells stably expressing shBeclin 1 were infected with Lenti-shp53 followed by western blot analyses (**d**), SA-β-gal staining (**e**), or colony formation assays (**f**). **g** H292 cells were subjected to IF staining. **h, i** H292 cells stably expressing shBeclin 1 (**h**) or shUSP5 (**i**) were subjected to nuclear-cytoplasmic fractionation assays. **j–l** A549 or H292 cells stably expressing shUSP5 were infected with lentivirus expressing Beclin 1 or Beclin 1$^{L184+187A}$. Cells were subjected to western blot analyses. (**j**), SA-β-gal staining (**k**), or colony formation assays (**l**). **m** A549 cells were subjected to IP-western blot analyses. **n, o** Purified recombinant His-Beclin 1 and His-Flag-p53 (**n**) or MDM2 (**o**) were subjected to in vitro protein binding assays. **p** A549 or H292 cells stably expressing shBeclin 1 were subjected to IP-western blot analyses. **q** Purified recombinant MDM2, His-Flag-p53 protein, and a dose of His-Beclin 1 (100, 200, or 400 ng) were subjected to in vitro protein binding assay. **r** A schematic diagram of Beclin 1-MDM2 interaction (upper panel). 293FT cells were transfected with MDM2-HA and Flag-Beclin 1 (FL) or an indicated deletion construct. Total cell lysates were subjected to IP-western blot analyses (lower panel). **s** A schematic diagram of MDM2-Beclin 1 interaction (left panel). 293FT cells were transfected with Flag-Beclin 1 and MDM2-HA (FL) or an indicated mutant construct. Total cell lysates were subjected to IP-western blot analyses (right panel). **t** 293FT cells transfected with p53-Myc, MDM2-HA, and Flag-Beclin 1 or an indicated mutant construct were treated with 20 μM MG132 before harvest. Total cell lysates were subjected to IP-western blot analyses. **u** A diagram depicts trimeric protein complexes among Beclin 1, p53, and MDM2. Experiments were performed three times independently (**a–t**). Data were presented as mean ± SD (**b, e, j, k**). Comparisons were performed with two-way (**b, e, j, k**) ANOVA with Tukey's test. Scale bar = 50 μm.

and in vitro (Fig. 5l). Ectopic expression of USP5, but not USP5$^{C195A}$, led to down-regulation of p53, resulting in increased colony formation (Supplementary Fig. S5j, k). In addition, activation of KRas promoted stabilization and activation of WT USP5, but not USP5$^{C195A}$ (Fig. 5m, n). Collectively, these results indicate that USP5 protein dimerization is critically important in Ras-induced cell proliferation and autophagy.

## USP5-Beclin 1-p53 axis is pivotal in *Kras*$^{G12D}$-driven lung tumor growth

We further investigated the significance of the KRas-USP5-Beclin 1-p53 pathway in *KRAS*-driven lung tumor growth. As shown in Fig. 6a, on the one hand, the expression of *KRAS*$^{G12V}$ significantly upregulated USP5, Beclin 1, or LC3-II expression and downregulated p62/SQSTM1 to promote autophagy. On the other hand, ablation of *USP5* or *BECN1* significantly upregulated p53 expression to induce cellular senescence and inhibit *KRAS*$^{G12V}$-induced cell growth (Fig. 6a–f).

As shown in Supplementary Fig. S6a–c, USP5 protein levels and enzymatic activities were markedly increased in the lung tissues derived from the *Kras*$^{G12D}$ mice, concomitant with the upregulation of Beclin 1 and decrease of p53 expression. Furthermore, the knockdown of *USP5* by the intranasal delivery of Lenti-shUSP5 in the *Kras*$^{G12D}$-driven lung tumorigenesis mouse model (Fig. 6g) led to dramatic suppression of lung tumor growth (Fig. 6h–j). To explore the basis as to why some residual tumors emerged from the *Kras*$^{G12D}$-shUSP5 mice, we examined the expression of USP5 in the residual tumors. IHC analyses showed that the expression of USP5 in the residual tumors derived from the *Kras*$^{G12D}$-shUSP5 was similar to the tumors derived from the control *Kras*$^{G12D}$ mice in the absence of shUSP5 (Supplementary Fig. S6c), suggesting that the presence of USP5 expression is most likely responsible for the residual tumor growth. Together, these results indicate that USP5 is critically important in *Kras*$^{G12D}$-driven lung tumorigenesis.

To further investigate the role of Beclin 1 in *Kras*$^{G12D/+}$-driven lung tumor progression. *Kras*$^{LSL-G12D/+}$;*Becn1*$^{+/+}$ or *Kras*$^{LSL-G12D/+}$;*Becn1*$^{flox/flox}$ mice were generated (Supplementary Fig. S6d, e). As shown in Fig. 6k–n, *Becn1* deficiency led to significant suppression of lung tumor growth. In contrast to *Kras*$^{G12D/+}$;*Becn1*$^{+/+}$ tumors, *Kras*$^{G12D/+}$;*Becn1*$^{-/-}$ tumors exhibited much-reduced autophagy as evidenced by accumulation of autophagy substrates p62/SQSTM1 and decrease of LC3 punctation (Fig. 6o). Furthermore, *Becn1* deficiency led to dramatically reduced cell proliferation in lung tumor tissue as evidenced by dramatically reduced Ki67 expression, concomitant with elevated expression of p53 and induction of senescence biomarkers such as PAI-1 and SA-β-gal activity (Fig. 6p and Supplementary Fig. S6f). Taken together, these results indicate that inhibition of USP5 or Beclin 1 leads to both induction of senescence and suppression of autophagy, resulting in significant blockage of *Kras*-driven lung tumor growth.

## Discussion

It has been well known that oncogenic Ras is unable to transform normal cells due to cellular senescence, and inactivation of senescence pathways involving p53 and RB is required for oncogenic Ras to promote transformation[1]. In the *Hras*$^{G12V}$-driven mouse mammary tumorigenesis model, chronic Ras activation upregulates the expression of the senescence-related proteins, including p53, p21, p16, p19$^{ARF}$, PAI-1, and SA-β-gal, concomitantly with the onset of proliferation arrest and inhibition of ductal elongation. Furthermore, in a *Kras*$^{G12Dint}$ colon cancer mouse model, *Kras*$^{G12Dint}$ mice develop widespread serrated hyperplasia but fail to progress into malignant tumors[47]. In this study, we demonstrate that *Kras*$^{G12D}$-mediated activation of the USP5-Beclin 1 axis is pivotal in overriding the intrinsic p53-imposed senescence burden in *KRAS*-driven lung tumorigenesis.

## Autophagy in *KRAS*-mediated tumorigenesis

The role of autophagy in *KRAS*-mediated tumorigenesis is still under debate. The notion that Beclin 1 can function as a tumor suppressor protein is supported by several well-known studies, including the observations that endogenous Beclin 1 protein expression is frequently low in human breast epithelial carcinoma cell lines and tissue, but is expressed ubiquitously at high levels in normal breast epithelia; *BECN1* is monoallelically lost in 40 to 75% of human breast, and ovarian cancers[6]. The *Becn1*$^{+/-}$ mice are prone to the development of liver and lung tumors and lymphomas[6,23,48,49]. However, large-scale genomic analyses of human cancers have failed to identify recurrent mutations in *BECN1*[50], implying that *BECN1* may not be a classical tumor suppressor gene in most human cancers[37]. Additionally, the *BECN1* allelic loss is confounded by its location adjacent to *BRCA1* on human chromosome 17q21, raising the possibility that loss of *BECN1* in human cancers may be associated with the loss of *BRCA1* in human breast and ovarian cancers[37]. Notably, in a *Palb2* loss mouse model for hereditary breast cancer, allelic loss of *Becn1* promotes p53 activation and reduces tumorigenesis[51], suggesting an intimate relation of p53 in Beclin 1-mediated regulation of tumorigenesis.

On the other hand, abundant evidence indicates that autophagy plays an important role in activating Ras-induced tumorigenesis[52], in which p53 is intimately involved. The *Kras*$^{G12V}$-driven salivary duct carcinoma progression is severely hindered by the deficiency of the autophagy gene *Atg7*[7]. Similarly, *Kras*$^{G12D}$-driven NSCLC is inhibited by *Atg7* deletion. *Atg7*-deficient tumors show prematurely induced p53 and proliferative arrest[53]. At later stages of tumorigenesis, *Atg7* deficiency causes p53 activation, proliferative defects, and reduced tumor burden[54].

In this study, we show that Beclin 1-mediated autophagy critically contributes to activated *Kras*-driven NSCLC in mice. Mechanistically, we demonstrate that activated KRas activates USP5, which in turn stabilizes Beclin 1 to induce autophagy. In the xenograft nude mice model, ectopic expression of Beclin 1 significantly facilitates tumor growth, whereas autophagy-deficient mutant Beclin 1$^{S90+93A}$ is unable to do so. Furthermore, activated *Kras*-driven NSCLC progression is inhibited by *Becn1* deficiency.

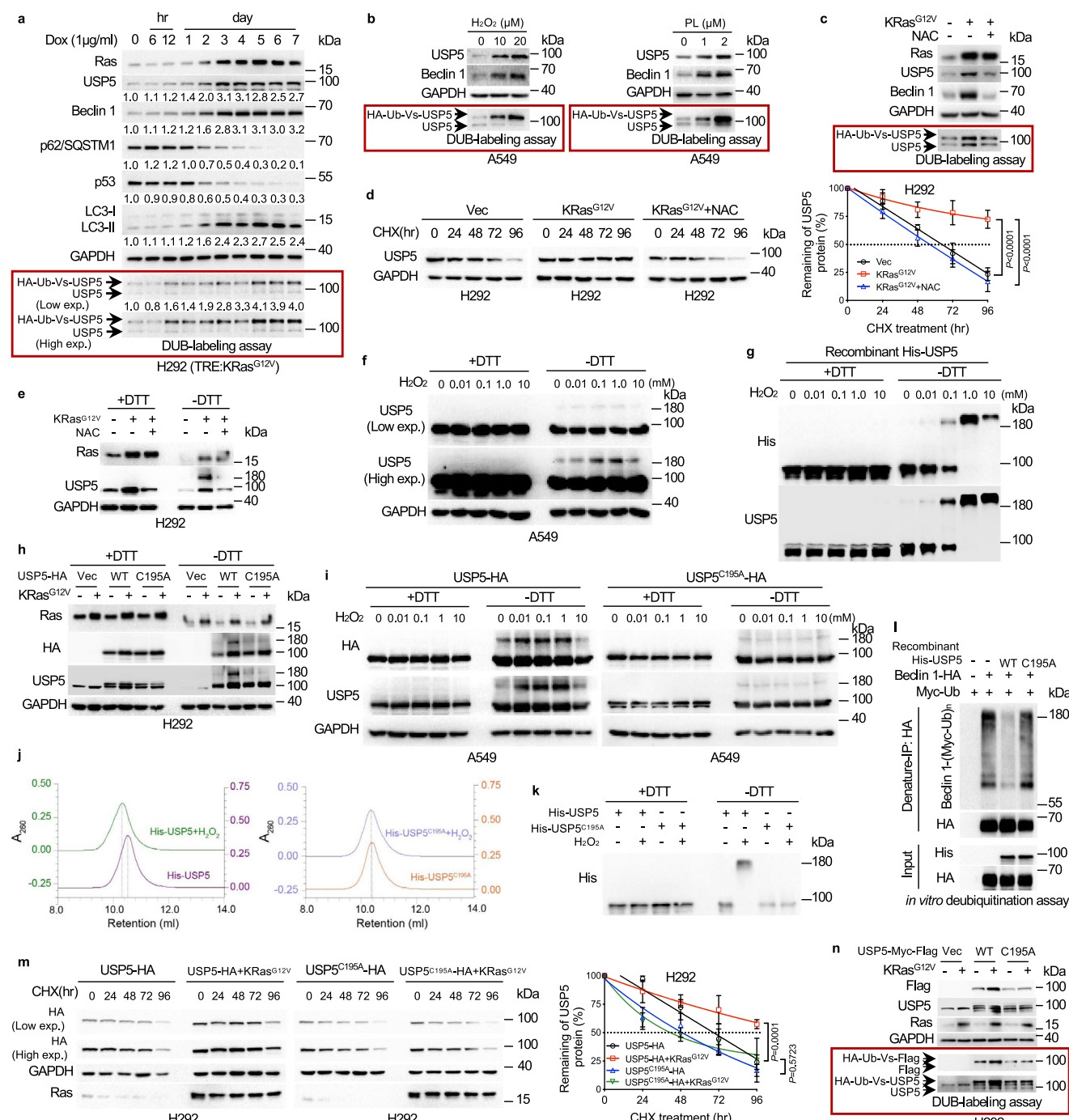

**Fig. 5 | *KRAS^G12V* promotes stabilization and activation of USP5 through ROS-induced disulfide-linked USP5 homodimerization involving C195. a** H292 cells harboring doxycycline-inducible *KRAS^G12V* were treated with doxycycline at the indicated time intervals and then subjected to western blot and the DUB-labeling assay for USP5 deubiquitination activity. **b** A549 cells treated with H₂O₂ (10 or 20 µM) or piperlongumine (PL, 1 or 2 µM) were subjected to western blot analyses and the DUB-labeling assays. **c** H292 cells stably expressing KRas^G12V were treated with 5 mM NAC for 24 h prior to western blot analyses or DUB-labeling assays. **d** H292 cells stably expressing KRas^G12V were treated in the presence or absence of NAC for 24 h, followed by USP5 protein half-life assays. **e** H292-*KRAS^G12V* cells were treated with 5 mM NAC for 24 h. Total cell lysates were subjected to SDS-PAGE with or without DTT (100 mM) in the loading buffer, followed by western blots. **f** A549 cells were treated with H₂O₂ at an indicated dose for 10 min prior to SDS-PAGE with or without DTT. **g** Purified recombinant His-USP5 protein in PBS was treated with

H₂O₂ at an indicated dose for 10 min prior to SDS-PAGE with or without DTT. **h** H292-*KRAS^G12V* cells stably expressing USP5-HA or USP5^C195A-HA were subjected to SDS-PAGE with or without DTT. **i** A549 cells stably expressing USP5-HA or USP5^C195A-HA were treated with H₂O₂ at an indicated dose for 10 min prior to SDS-PAGE with or without DTT. **j, k** Purified recombinant His-USP5 or His-USP5^C195A protein in PBS (1 mg/200 µL) were treated with 1 mM H₂O₂ for 10 min followed by HiTrapQ FPLC ion-exchange chromatography. Gel filtration of USP5 dimers or monomers showed distinct peaks (**j**), verified by western blot analyses (**k**). **l** Ubiquitinated Beclin 1-HA expressed in 293FT cells was purified by denature-IP and incubated with recombinant His-USP5 or His-USP5^C195A. **m, n** H292-USP5-HA or H292-USP5^C195A-HA cells stably expressing *KRAS^G12V* were subjected to USP5 or USP5^C195A protein half-life (**m**) or DUB-labeling assays (**n**). Experiments were performed three times independently (**a–n**). Data were presented as mean ± SD and comparisons were performed with two-way ANOVA with Tukey's test (**d, m**).

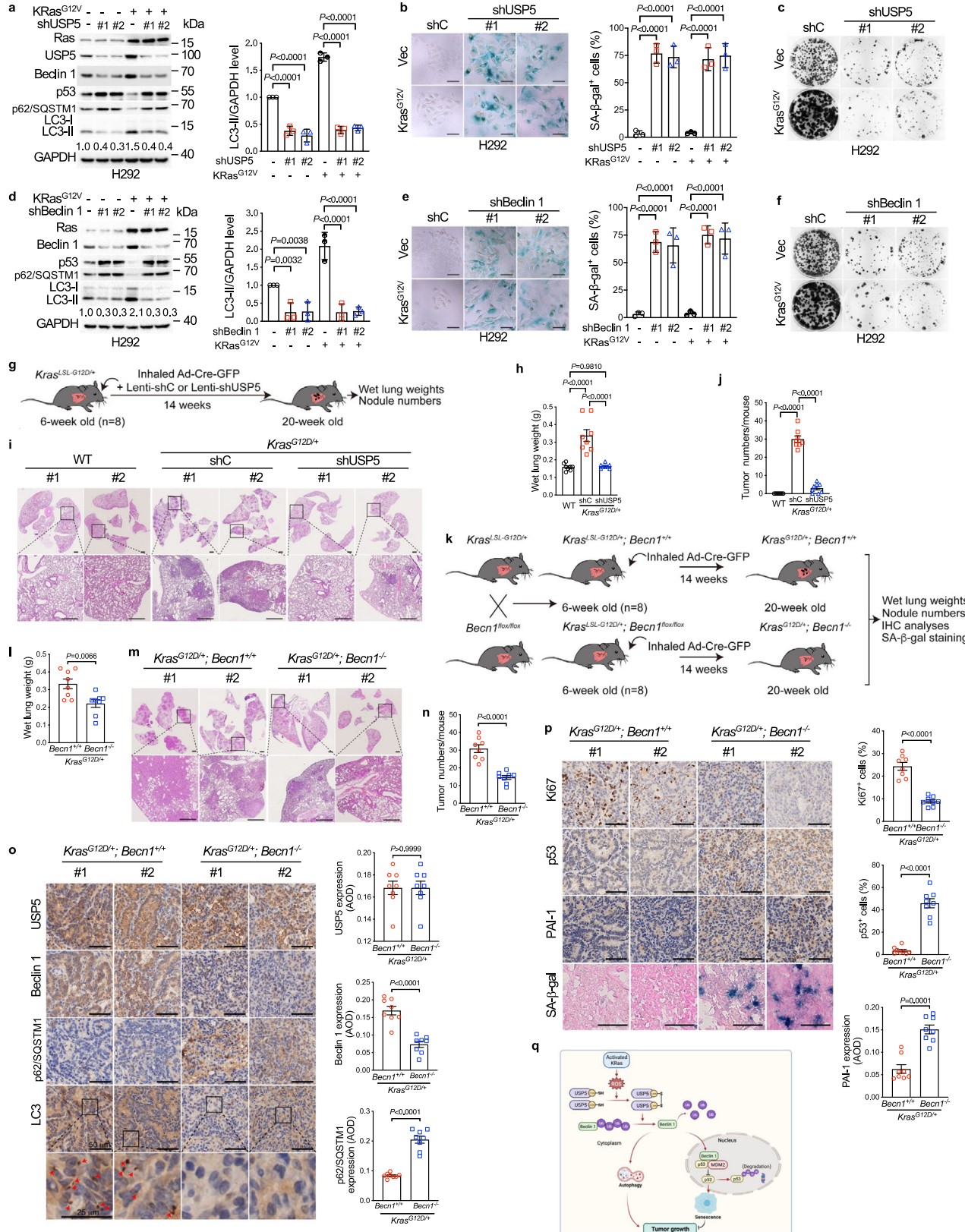

## Nuclear Beclin 1 on the modulation of p53

Beclin 1 can be localized in both cytoplasm and nucleus. The cytoplasmic Beclin 1 functions to form the Beclin 1-Vps34-Vps15-ATG14 protein complex to promote autophagy initiation and autophagosome formation. Nuclear Beclin 1 is shown to play a role in the repair of ionizing radiation (IR)-induced DNA double-strand break[55].

Unexpectedly, in this study, we found that nuclear Beclin 1 forms stable Beclin 1-p53-MDM2 trimeric complexes to modulate p53 protein stability.

The BH3 domain of Beclin 1 (AA 88-150), essential for autophagy, can interact with Bcl-2/Bcl-X$_L$ to block Beclin 1 interaction with Vps34, thereby preventing autophagy-inducing class III PI3K core complex

**Fig. 6 | Depletion of *USP5* or *Becn1* induces both p53-dependent senescence and autophagy inhibition resulting in suppression of *Kras*-driven lung tumor growth. a–f** H292-*KRAS*^G12V^ cells stably expressing shUSP5 or shBeclin 1 were subjected to western blot analyses (**a, d**), SA-β-gal staining (**b, e**), or colony formation assays (**c, f**). Quantification of the LC3-II/GAPDH ratio was shown. Three independent experiments were performed (**a–f**). Representative photos were shown and the quantification of SA-β-gal staining positive cells was presented as mean ± SD. Comparisons were performed with two-way ANOVA with Tukey's test (**a, b, d, e**). Scale bar = 50 μm. **g–j** Silencing of *USP5* inhibits *Kras*^G12D^-driven lung tumor growth. A diagram depicts an experimental design to examine the effects of *USP5* silencing on lung tumor growth in *Kras*^LSL-G12D/+^ mice (**g**); Results of wet lung weights (**h**); representative histology (H&E) and tumor burden (**i**); and tumor numbers (**j**) were shown. *n* = 8/group (**h, j**). **k–p** *Becn1* deficiency suppresses *Kras*^G12D^-driven lung

tumor growth through increased senescence burden and reduced autophagy. A diagram depicts an experimental design to examine the effects of *Becn1* deficiency on lung tumor growth in *Kras*^LSL-G12D/+^ mice (**k**); Results of wet lung weights (**l**); representative H&E of lungs and tumor burden (**m**); tumor numbers (**n**); representative IHC images and SA-β-gal staining, as well as the corresponding quantification of USP5, Beclin 1, LC3, p62/SQSTM1, Ki67, p53 and PAI-1 (**o, p**) were shown. Red arrows point to LC3 aggregates. *n* = 8/group (**l, n–p**). Data were presented as mean ± SEM (**h, j, l, n–p**). Comparisons were performed with two-way ANOVA with Tukey's test (**h, j**) and two-tailed Student's *t*-test (**l, n–p**). Scale bar = 50 μm. **q** A model depicts that oncogenic KRas activates USP5 to regulate Beclin 1 and p53 in promoting autophagy and overriding cellular senescence. Notably, cytoplasmic Beclin 1 promotes autophagy, whereas nuclear Beclin 1 inhibits the p53 pathway.

fromation[23]. Our results indicate that the Beclin 1 BH3 domain is also the binding site for p53, raising an interesting possibility that cytoplasmic p53 binds to the BH3 domain and interferes with Vps34 interaction with Beclin 1, thereby inhibiting autophagy. Indeed, it has been reported that cytoplasmic, but not nuclear p53 can repress autophagy[56].

In this study, we show that Beclin 1 CCD domain (AA 150-244) binds to the segment (AA 154-221) of MDM2 and promotes MDM2-p53 interaction resulting in accelerated proteasomal degradation of p53 protein. Thus, nuclear Beclin 1 can function as an adapter to facilitate trimeric protein complex formation to destabilize p53. Interestingly, Beclin 1 can also modulate p53 protein stability by regulating the activities of USP10/USP13, which can deubiquitinate p53[57]. Thus, Beclin 1 can modulate p53 protein stability via different means, likely in a context-dependent manner.

### Autophagy and cellular senescence
Autophagy is implicated in a variety of biological processes, including survival, cancer development, neuronal degeneration diseases, aging, and cellular senescence[58], although the precise role of autophagy in senescence remains in debate. Knockout of *Atg7* in the muscle satellite cells leads to increased senescent satellite cells and the loss of stemness[59]. Recent studies reported that autophagy can promote senescence through the degradation of the nuclear Lamin B1, chromatin, and SIRT1[60,61]. In this study, we show that knockdown of *BECN1* leads to p53 stabilization resulting in cellular senescence and that oncogenic Ras elevates nuclear Beclin 1 expression to destabilize p53 protein to overcome p53-imposed cellular senescence, independent of its pro-autophagic function. This premise is corroborated by the observations that Beclin 1^L184+187A^, which is unable to promote autophagy, can rescue *USP5* ablation-induced upregulation of p53 expression and senescence.

### USP5-mediated modulation of Beclin 1
Beclin 1 protein stability is critically regulated by the ubiquitin-proteasome system. Several E3 ubiquitin ligases and DUBs (USPs) have been reported to regulate Beclin 1 protein stability[62]. For instance, RNF216 or SKP2 promotes K48-ubiquitination and destabilization of Beclin 1 to inhibit autophagy in response to pathogen or viral infection. Beclin 1 undergoes K48-ubiquitination and is destabilized by Cul3-KLHL20 to reduce autophagy upon prolonged starvation[62,63]. Notably, NEDD4 can promote ubiquitination on K6, K11, K63, or K27 to impact Beclin 1 protein stability[62]. On the other hand, Ataxin 3 deubiquitinates K48 and stabilizes Beclin 1 to promote autophagy. USP19 deubiquitinates K11 and stabilizes Beclin 1[30]. In addition, USP10 or USP13 can deubiquitinate and stabilize Beclin 1 in promoting autophagy[57]. In this study, we discovered that USP5 is a bona fide deubiquitinase to bind to and stabilizes Beclin 1 protein in controlling both Beclin 1-dependent autophagy and p53-dependent senescence. Importantly, USP5 is indispensable for *Kras*-mediated NSCLC progression. Consistent with

this notion, the knockdown of *USP5* dramatically inhibits *Kras*-induced lung tumor growth.

### Oncogenic *KRAS* signaling and ROS-mediated disulfide bond formation of USP5
ROS plays an essential role in KRas-mediated tumorigenesis[64]. ROS can promote intramolecular or intermolecular disulfide bond formation, leading to dimer formation to affect the stabilization or activity of downstream target proteins, such as KEAP1 or PTEN[21]. In this study, we show that activated KRas elevates cellular ROS, which in turn induces the formation of USP5 homodimers through the intermolecular C195 disulfide bond, resulting in increased deubiquitinating enzymatic activity of USP5 and the accumulation of USP5 protein. Notably, both KRas and ROS inducers (H₂O₂ or piperlongumine) elevate USP5 activity and USP5 protein expression in a ROS-dependent manner, the latter of which greatly contributes to the upregulation of USP5 activity. Importantly, KRas can stabilize and stimulate the activity of WT USP5, but not USP5^C195A^. Furthermore, WT USP5, but not USP5^C195A^, can deubiquitinate and stabilize Beclin 1, consequently promoting autophagy and colony formation. Collectively, these results demonstrate that oncogenic KRas activates the USP5-Beclin 1 axis to promote autophagy and to override intrinsic p53-imposed cellular senescence, which is pivotal in *KRAS*-driven lung tumorigenesis (Fig. 6q).

### Therapeutic targets for *KRAS* cancers retaining wild-type p53
It is well documented that various cancers with activation of a driver oncogene often retain a WT *TP53* allele (Supplementary Fig. S6g). For instance, in human NSCLC, the *KRAS* is frequently mutated (23%) with 63% WT *TP53* alleles. In human bowel cancers, the *KRAS* is frequently mutated (38%) with 41% WT *TP53* alleles. In this study, we show that silencing of *USP5* dramatically inhibits autophagy and induces p53-dependent senescence burden, resulting in complete inhibition of *p53*+ cancer cell growth and xenograft tumor formation. Similar results are obtained by pharmacological inhibition of USP5 using WP1130. Importantly, silencing of *USP5* dramatically blocks *Kras*^G12D^-driven tumor growth in mice models, in which silencing of *USP5*-induced p53-dependent senescence burden plays a key role. Thus, targeting USP5 to reduce autophagy and impose senescence burden may be a potential therapeutic strategy for cancer treatment.

## Methods
### Ethics statement and mouse models
All the mouse strains were kept in standard, infection-free housing conditions, with 12 h light:12 h dark cycles and three to five mice per cage. Animals were housed in a pathogen-free barrier environment throughout the study. All animal experiments in this study were approved by the Institutional Animal Care and Use Committee of Sichuan University (IACUC), and the operating procedures were carried out in accordance with the guidelines formulated by China Council on Animal Care.

For in vivo tumor progression, Antares2-expressing A549 cells ($2 \times 10^5$) or their derivatives were subcutaneously transplanted into the right front flanks of 6-week-old female nude mice ($n = 5$/group) (Model Animal Research Center of Nanjing University, Nanjing, China). Tumor size was measured with a caliper every 2 days, and tumor volume was calculated by width$^2 \times$ length $\times 1/2$. The maximal tumor size permitted by the IACUC of Sichuan University is 20 mm at the largest diameter in mice. About 0.1 mM Diphenylterazine (DTZ, MCE) (the substrates of antares2) in 100 μL PBS were injected into tumor regions in anesthetized mice using tribromoethanol (250 mg/kg intraperitoneally (i.p.)). Mice were recovered on heat pads for 5 min and bioluminescence was subsequently imaged using a Caliper IVIS Lumina (Perkin Elmer, Waltham, USA). The images are processed using the Living Image 4.5.5 software.

For transgenic mice, the *Becn1* conditional knockout mice (*Becn1*-flox/flox) were obtained from Shanghai Model Organisms Center (Shanghai, China). *Kras*$^{LSL-G12D/+}$ mice (provided by Dr. Chong Chen of Sichuan University). The compound *Kras*$^{LSL-G12D/+}$;*Becn1*flox/flox mice were identified by genotyping using specific primers listed in Supplementary Table S1. Six-week-old animals were anesthetized before intratracheal delivery to each mouse ($n = 8$) with a mixture of adenoviruses ($2.5 \times 10^7$ IFU) expressing Cre-recombinase-GFP (Ad-Cre-GFP)[65] and lentivirus ($1 \times 10^5$ IFU) expressing shUSP5 (Lenti-shUSP5) or Lenti-shC in 100 μL serum-free MEM. Mice were grown for another 14 weeks before sacrifice.

## Cell culture and reagents
NCI−H292 (CRL-1848) was obtained from ATCC (Manassas, VA, USA). A549 (BNCC337696) was obtained from BeNa Culture Collection (Beijing, China). SK-LU-1 (SNL-492) and NCI-H358 (SNL-392) were obtained from Sunncell Biotech (Wuhan, China). NCI-H1299 (CL-0165) and H1975 (CL-0298) cells were obtained from Procell Life Science& Technology (Wuhan, China). HEK293FT (R70007) cells were from Thermo Fisher Scientific (Waltham, MA, USA).

A549, HEK293FT, SK-LU-1, and NCI−H1299 cells were cultured in DMEM medium (GIBCO, Rockville, MD, USA) supplemented with 10% fetal bovine serum (Hyclone, Logan, UT, USA) and penicillin (100 U/ mL)/ streptomycin (100 μg/mL) (Hyclone). NCI−H292, NCI−H358, and NCI−H1975 cells were cultured with 10% FBS supplemented with penicillin/streptomycin in RPMI 1640 medium (GIBCO). Cells were maintained at 37 °C in a humidified 5% $CO_2$ incubator. MG132 (S2619), cycloheximide (S7418), Piperlongumine (S7551), Puromycin (S7417), or WP1130 (S2243) were purchased from Selleck Chemicals (Houston, USA). Blasticidin (203351), polybrene (TR-1003), and anti-FLAG® M2 affinity gel (A2220) were purchased from Sigma-Aldrich (St. Louis, USA). Pierce™ Anti-HA magnetic beads (88836) were purchased from Thermo Fisher Scientific. Diphenylterazine (HY-111382) was purchased from MCE (Shanghai, China). *N*-acetylcysteine (ST1546) was purchased from Beyotime (Shanghai, China).

## Plasmids, lentiviral infection, and RNA interference
The plasmids used in generating recombinant lentiviruses, including pLenti-M3-USP5-Myc-Flag, pLenti-M3-USP5$^{W209A}$-Myc-Flag, pLVX-USP5-HA, pLVX-USP5$^{C195A}$-HA, pLVX-Beclin 1, pLVX-Beclin 1$^{S90+93A}$, pLVX-Beclin 1$^{L184+187A}$, pLVX-Beclin 1-HA, pLVX-Beclin 1$^{L184+187A}$-HA, pLVX-Antares2, pLenti-CMV-RFP-GFP-LC3, and pCW-KRas$^{G12V}$ were constructed in this study and verified by direct DNA sequencing. Lentiviral-based short hairpin RNAs (shRNAs) specific for USP5, Beclin 1, p53, or the scramble controls, were cloned into pLKO.1-TRC. Targeted sequences were listed in Supplementary Table S1. The human deubiquitinating (DUB) Library[66] consists of 85 expressing plasmids (pCMV6Entry), in which the expressing plasmids encoding 45 USP genes were listed in Supplementary Table S2.

## Prokaryotic expression, purification, and FPLC analysis of USP5
For prokaryotic expression and purification of USP5, the coding sequence fragment was cloned into a pHIS2 vector. Recombinant USP5

or USP5$^{C195A}$ proteins were produced in *E. coli* BL21(DE3) cells induced by IPTG, and affinity-purified by HisTrap HP Histag protein purification column (17524801, Cytiva, Uppsala, Sweden). Final purification of USP5 proteins was achieved with a Superdex 200 increase 10/300 GL column (28990944, Cytiva) on BioLogic DuoFlow Pathfinder 20 System (7602258, Bio-Rad). For FPLC analytical analyses of USP5, 1 mg purified His-USP5 or His-USP5$^{C195A}$ protein in 200 μL buffer was treated with 1 mM $H_2O_2$ for 10 min on ice, which was then loaded onto the Superdex 200 increase 10/300 GL column at 4 °C. The peaks of USP5 protein dimers and monomers were monitored and collected, followed by verification using western blot analyses.

## Western blot, immunofluorescence (IF), and immunohistochemistry (IHC) analyses
Western blot, IHC, and IF analyses were performed essentially as described in ref. 38. Briefly, for western blot analyses, cells were washed twice with PBS and lysed with EBC250 buffer (250 mM NaCl, 25 mM Tris-HCl, pH 7.4, 0.5% NP-40, and 50 mM NaF) supplemented with protease inhibitor cocktail (B14001, Selleck Chemicals). Equal amounts of total protein were fractionated by SDS/PAGE and transferred to the PVDF membrane. Non-specific binding was blocked in 4% nonfat dry milk diluted in TBS supplemented with 0.1% Tween and membranes were incubated with primary antibody and HRP-conjugated secondary antibody for subsequent detection by chemiluminescence (Bio-Rad). Images were analyzed using Image Lab Software 5.1. Antibodies specific for USP5 (sc-390943, 1:200) and p53 (sc-126, 1:200 or sc-6243, 1:200) were purchased from Santa Cruz Biotechnology (CA, USA); antibodies for Beclin 1 (#3495, 1:1000 or #4122, 1:1000), LC3B (#2775, 1:1000), p62/SQSTM1 (#5114, 1:1000), HA (#2367, 1:1000), Flag (#14793, 1:1000), Ras (#8955, 1:1000), PARP (#9532, 1:1000), and Caspase-3 (#9662, 1:1000) were purchased from Cell Signaling Technology (Danvers, MA, USA); antibodies for p21 (CY5543, 1:1000), MDM2 (CY3612, 1:1000) and GAPDH (AB0037, 1:5000) were purchased from Abways (Shanghai, China); antibody specific for His (230001, 1:1000) was purchased from Zen BioScience (Chengdu, China).

For IHC analyses, human lung adenocarcinoma tissue microarrays (TMA) (HLugA180Su04) were obtained from Outdo Biotech (Shanghai, China). Paraffin-embedded tumors or lung samples were sliced into 5 μm thickness. Tissue sections were rehydrated through a decreasing ethanol gradient, and treated by boiling in citrate buffer (pH 6.0) or Tris-EDTA (pH 9.0) for antigen retrieval. Endogenous peroxidases were blocked using 0.3% $H_2O_2$. After blocking with 5% BSA, the sections were incubated with primary antibody and followed by horseradish peroxidase-conjugated secondary antibody. The sections were subsequently stained with a DAB Detection Kit (ZLI-9018, ZSGB-BIO). Antibodies used in IHC for USP5 (ab244290, 1:400), Beclin 1 (ab62557, 1:400), LC3B (ab48394, 1:400), and p62/SQSTM1 (ab56416, 1:400) were purchased from Abcam (Cambridge, MA, USA); antibody for PAI-1 (#11907, 1:800) was purchased from Cell Signaling Technology; antibody for human p53 (sc-126, 1:200) was purchased from Santa Cruz Biotechnology; antibody for mouse p53 (NCL-L-p53-CM5p, 1:400) was purchased from Leica Biosystems (Newcastle, UK). For quantitative analysis, tissue slides were scanned through NanoZoomer (Hamamatsu, Japan), and the scanned images were subjected to analyzing average optical density (AOD)[38] using QuPath[67].

For IF analyses, cells were fixed with 4% paraformaldehyde for 15 min at room temperature, permeabilized with 0.1% Triton-100 for 15 min, blocked with 5% BSA for 1 h, and stained with specific primary antibodies followed by corresponding secondary antibodies. Antibodies used in IF for USP5 (sc-390943, 1:100) or p53 (sc-126, 1:100) were purchased from Santa Cruz Biotechnology; antibody for Beclin 1 was purchased from Abcam; antibody for HA (#2367, 1:400) was from Cell Signaling Technology. Rhodamine (TRITC)-conjugated AffiniPure Donkey Anti-Rabbit IgG (#711-025-152, 1:160) and Fluorescein (FITC)-

conjugated AffiniPure Donkey Anti-Mouse IgG (#715-095-150, 1:160) were purchased from Jackson Immuno Research (PA, USA). For RFP-GFP-LC3 puncta analyses, cells stably expressing RFP-GFP-LC3 were stained with Hoechst and then were visualized using Leica Confocal microscope TCS SP5 II to image the GFP- and RFP- LC3 puncta, respectively. Fluorescent micrographs of the GFP- and RFP-channel were further analyzed to obtain numbers of subcellular GFP- or RFP-LC3 puncta per cell using the QuPath[67]. Five randomly selected ROIs (regions of interest) and over 200 cells were analyzed in each experiment.

### Screen for DUB regulation of autophagy
The 293FT cells were infected with the recombinant lentivirus expressing pLenti-CMV-RFP-GFP-LC3 and selected for stable cells displaying uniform and evenly distributed red/green fluorescence by flow cytometer (FACS AriaIII, BD), These cells, named 293FT-RFP-GFP-LC3 reporter, were used for transient expression of each of the USP genes (USP1-USP53) from the DUB library[66]. Seventy-two hours post-transfection, cells were subjected to FACS analyses for the changes in GFP fluorescence intensity. The EBSS-treated reporter cells were used in parallel as a positive control to reflect increased autophagy.

### Quantitative PCR (qPCR) analysis
Total RNA was extracted using NucleoSpin® RNA Plus kit (740984, MACHEREY-NAGE), followed by reverse transcription using ReverTra Ace qPCR RT Master Mix (FSQ-201, TOYOBO). qPCR analyses were performed as previously described in ref. 38. The specific primer sequences for qPCR were listed in Supplementary Table S1.

### DUB-labeling assay, ubiquitination assay, and protein–protein interaction assays
The measurement of USP5 enzymatic activity was performed using the HA-Ub-Vs probe (Boston Biochem)[68]. Briefly, cells were lysed in DUB assay buffer (50 mM Tris-HCl pH 7.2, 5 mM $MgCl_2$, 250 mM sucrose, protease inhibitor cocktail (Selleck), 1 mM NaF, and 1 mM PMSF) for 10 minutes at 4 °C, followed by brief sonication. The lysates were centrifuged and the supernatant (20 μg) was incubated with 200 nM HA-Ub-Vs in a total volume of 40 μL of DUB assay buffer at 37 °C for 1 h, followed by western blot.

For in vivo ubiquitination assay, HEK293FT cells were transfected with indicated expression plasmids or infected with indicated recombinant lentivirus. Cells were incubated with 20 μM MG132 for 6 h prior to collection if needed. The collected cells were lysed in a denaturing lysis buffer (50 mM Tris-HCl pH 7.4, 150 mM NaCl, 1% NP-40, and 1% SDS with protease and phosphatase inhibitors). Cell lysates were boiled for 10 min, diluted ten times in lysis buffer without SDS, and subjected to immunoprecipitation with anti-HA agarose beads followed by western blot[69].

For in vitro ubiquitination assay, HEK293FT cells expressing Beclin 1-HA and Myc-Ub were incubated with 20 μM MG132 for 6 h and then were lysed in a denaturing lysis buffer. Cell lysates were boiled for 10 min and diluted ten times in lysis buffer without SDS, followed by purification using anti-HA agarose beads. The same amount of beads was incubated with 0.01 μg/μL recombinant His-USP5 or His-USP5$^{CI195A}$ in the deubiquitination reaction buffer (50 mM Tris-HCl pH 7.4, 150 mM NaCl, 5 mM $MgCl_2$, and 10 mM DTT) at 37 °C for 1 h in the presence of ATP (1 μM).

For endogenous co-immunoprecipitation analyses[66], cells were lysed with IP lysis buffer (20 mM Tris-HCl, 125 mM NaCl, 5 mM $MgCl_2$, 0.2 mM EDTA, 12% Glycerol, and 0.25% NP-40). The lysates were incubated with primary antibodies for 6 h at 4 °C, and then 25 μl of protein A/G beads were added for an additional 2 h of incubation. For exogenous Co-IP, anti-HA/Flag beads were added to equal amounts of total protein and incubated for 6 h. Beads were washed three times using IP wash buffer (20 mM Tris-HCl, 125 mM NaCl, 5 mM $MgCl_2$,

0.2 mM EDTA, and 0.1% NP-40), and then boiled for 10 min before SDS-PAGE and western blot.

For in vitro binding assay, recombinant human His-Beclin 1 protein (NBP2-22710) was purchased from Novus Biologicals, and recombinant human MDM2 protein (E3-204) and recombinant human His6-Flag-p53 protein (SP-452) were purchased from Boston Biochem. Recombinant proteins (400 ng each) were incubated in binding buffer (20 mM Tris-HCl pH 7.8, 125 mM NaCl, 5 mM $MgCl_2$, 0.2 mM EDTA, 12% glycerol, 0.25% NP-40, 1 mM PMSF, protease and phosphatase inhibitors) for 6 h, and then were incubated with anti-Flag agarose beads or anti-Beclin 1 antibody followed by Protein A beads (Santa Cruz, sc-2001) to purify. After washing three times in binding buffer, proteins were eluted by mixing with loading buffer, boiled for 5 min, and subjected to western blot assay.

### Colony formation and senescence-associated β-galactosidase (SA-β-gal) assays
A549 (500) or H292 (1000) cells were seeded in a 6-well plate and cultured for 20 days. The colonies were fixed for 15 min with cold methyl alcohol, stained with 0.1% crystal violet for 15 min at room temperature, and photographed. For the SA-β-gal assay in vitro[70], cells were fixed, stained with X-Gal solution, and washed in PBS. Representative pictures were taken under bright-field microscopy. To identify senescent cells in vivo, frozen lung tissues derived from mouse models were cut into 10-μm-thick sections, followed by β-gal and eosin staining.

### Statistics and reproducibility
Data from three independent experiments in vitro were presented as mean ± SD, and data from animal experiments were presented as mean ± SEM. Two-tailed unpaired Student's $t$-test was used for comparing two groups of data. One/two-way ANOVA with Tukey's or Bonferroni's multiple-comparison test was used to compare multiple groups of data. $P$ values of less than 0.05 were considered significant.

### Reporting summary
Further information on research design is available in the Nature Portfolio Reporting Summary linked to this article.

## Data availability
The structures of the USP5 protein crystal structure (accession code 3IHP [10.2210/pdb3ihp/pdb]) were obtained from Protein Data Bank (https://www.rcsb.org/). The KM plotter lung cancer dataset was obtained from http://kmplot.com/analysis. All data generated or analyzed during this study are included in this article and its Supplementary Information files. The uncropped gel or blot figures and original data underlying Figs. 1–6 and Supplementary Figs. S1–S7 are provided as a Source Data file. Source data are provided in this article. All the other data are available within the article and its Supplementary Information. Source data are provided with this paper.

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

## Acknowledgements

We are grateful to Drs. Chong Chen and Yu Liu (State Key Laboratory of Biotherapy, West China Hospital, Sichuan University, China) for *Kras^LSL-G12D/+* mice. Dr. Zhonghan Li (Sichuan University, China) for the Antares2 plasmid and Dr. Jianbo Yue (City University of Hong Kong, China) for the pLenti-CMV-RFP-GFP-LC3 plasmid. We thank Dr. Dongxi Xiang (Shanghai Jiao-tong University School of Medicine, China) for the suggestions and members of the Z.-X.J.X. laboratory for stimulating discussions during this study. This work was supported by the National Key R&D Program of China (2018YFC2000100 and 2022YFA1103700 to Z.-X.J.X.), National Natural Science Foundation of China (81830108 to Z.-X.J.X., 82103167 to Y.W., and 82188101 to J.Y.), the Strategic Priority Research Program of the Chinese Academy of Sciences (XDB39030200 to J.Y.), Shanghai Municipal Science and Technology Major Project (2019SHZDZX02 to J.Y.), and China Postdoctoral Science Foundation (2021M702363 to Y.W.).

## Author contributions

Z.-X.J.X., Y.W., and Juan.L. conceived and designed the research. Juan.L. and Y.W. performed most of the biochemical and molecular experiments, with assistance from Yue.L., Jing.L., Y.P., W.L., W.Y., and Q.H. Animal experiments and IHC analyses were performed by Juan.L., Yue.L., Y.W., and Z.Z. Computational biology analyses were performed by Y.C. and Yang.L. Y.Y., Y.Z., L.Z., and J.Y. contributed to the data discussion. Z.-X.J.X., Y.W., Juan.L., and J.Y. wrote the manuscript. All the authors read and approved the manuscript.

## Competing interests

The authors declare no competing interests.
