## [Peer Review File · Nature Communications]

USP5-Beclin 1 axis overrides p53-dependent senescence and drives *Kras*-induced tumorigenicityREVIEWER COMMENTS

Reviewer #1 (Remarks to the Author):

This is an excellent and thoughtful manuscript that characterizes the relationship between autophagy and senescence in lung cancer development. The work further identifies interaction(s) between USP5, Beclin and p53 in the regulation of autophagy and senescence and raises the possibility that USP5 influences clinical outcomes in lung cancer patients. The studies presented utilize appropriate technical approaches, particularly cells with expression of recombinant/mutant proteins and shRNA silencing to interrogate the putative relationships and impact on senescence and clonogenic survival. These latter data are particularly robust and convincing of the tumor growth repressive impact of senescence.

A few deficiencies that likely could be readily corrected are indicated below.

The Western blots should be accompanied by quantification of the protein bands. This is a problem, particularly in the case of LC3I to II conversion, where this indication of autophagy does not always appear to reflect the modulation of autophagy that the authors propose.

A critical finding is that KRAS/G12D up regulates USP5 to promote autophagy, which prevents the cells from entering into senescence, and facilitates transformation. However, it appears that only a single time point is presented in support of this conclusion in Figure 5. This fundamental observation requires substantial confirmation.

The authors refer to decreased p62 on line 221, but there is no reference to where this information could be found in a figure. It is further unclear why p62 levels are not assessed routinely, for instance in Figure 4j. It should be noted that p62 is routinely referred to as SQSTM1.

Minor issue: Lines 40-41. The observations relating to the LC3 puncta could be explained in somewhat greater detail.

The Discussion section is somewhat rambling and difficult to follow and could be tightened up, perhaps by having sections with subheadings. The authors could also consider addressing previous literature where the relationship between autophagy and senescence did not follow the same pattern as in the current work; that is, autophagy may have been reported to promote senescence or be required for senescence, rather than preventing senescence.

Reviewer #2 (Remarks to the Author):

In this study, the authors investigate the role of USP5-Beclin1 axis in KRAS driven tumor progression. This topic is novel and interesting and the methodology is sound. The authors found that knockdown of USP5 or Beclin1 increase senescence and suppressed tumor growth in mice. Mechanistically, USP5 is stabilized by dimerization that can be induced by ROS. Stabilized USP5 improves Beclin1 protein stability, leading to degradation of p53. Knockdown of USP5 or Beclin1 promotes p53-dependent senescence and inhibits autophagy, resulting in suppression of KRAS driven NSCLC growth. This study opens a new window into developing strategies to target USP5 for cancer therapy. There are, however, some weak points in the paper that need to be clarified to obtain sufficient support for some of the conclusions.

Main comments:

1. In Fig1e, it seems that knockdown of USP5 by two shRNAs is toxic for cell colony growth, however, the selectively specific inhibitor of USP5, WP1130 shows similar effect on Beclin1, p53 and p21 as shUSP5, but it does not result in similar suppression of cell growth. Are there off target effects of the shUSP5s or is the inhibitor ineffective in inhibiting USP5?

2. The authors highlight the role of USP5-Beclin1 axis for overriding p53-dependent senescence in KRAS-driven tumors. However, the cell models used here are A549 (KRAS-G12S) and H292 (KRAS

WT). The results of manipulating USP-Beclin1 axis in both cell lines are the same. Does this mean that the USP-Beclin1 axis functions in lung cancer cell regardless of KRAS mutation? If so, please explain how.

3. Similarly, it would be important to define the role of the axis in p53 WT and p53-mutant/ko tumor cells, regardless of KRAS mutation.

4. The authors show nicely in vivo ubiquitination assay results in Fig3. Please explain whether this assay was performed under denaturing conditions. From the Methods in the manuscript, it seems the assay was performed with a normal Co-IP protocol, and not under denaturing conditions. As the results from ubiquitination assay are key parts of mechanism work, it will be important to explain and motivate in detail that how the in vivo ubiquitination assay was performed.

5. Then, an in vitro ubiquitination assay related to the Fig 3 should be performed.

6. The Co-IP bands in Fig4m are distinctly and clearly displayed. The bands for p53 and IgG should be very similar in size. Thus, how did you technically go about separating them?

7. In Fig5a and c, overexpression of KRAS-G12V upregulated the expression of USP5. By comparing the ratio between HA-Ub-Vs-USP5 and USP5, it seems that the increased USP5 activity was caused by increased protein expression of USP5. Hence, it seems a stretch to conclude that "KRAS-G12V significantly increased USP5 deubiquitinating enzymatic activity". In Fig5b, ROS inducers, H₂O₂, and PL clearly increased USP5 activity by comparing the ratio between HA-Ub-Vs-USP5 and USP5. The addition of NAC can't change the USP5 ratio. The authors should explain this further. It would also be informative if the authors include an experiment where cells that do not overexpress KRAS-G12V are incubated with NAC.

8. In Fig5I, shortly exposed bands in USP5-HA+KRAS-G12V sample should be added. If possible, significance levels should be analyzed in quantifications of CHX treatment results in Fig1b and c, Fig3a, Fig5d and I, and FigS4d.

9. The authors propose that USP5 and Beclin1 can regulate p53 stability and promote MDM2-mediated p53 degradation. A ubiquitination assay for p53 should be performed to directly confirm MDM2-mediated proteasomal-dependent degradation of p53, for example, by depleting USP5 or Beclin1.

10. It is known that inhibition of USP5 induces apoptosis. Considering the role of p53 in cell apoptosis, how do authors exclude the possibility that the suppression of tumor growth by depletion of USP5 was not caused by increased apoptosis?

Reviewer #3 (Remarks to the Author):

The manuscript by Juan Li et al discovered that USP5-Beclin 1 axis is critically involved in lung cancer cell senescence and tumorigenesis. Using a series of elegant experiments including cell biology, mechanistic studies, animal models and patient samples, the authors reported that USP5 stabilizes Beclin 1, which regulates p53 stability and senescence. Further, they showed that oncogenic Ras modulates USP5 via ROS, and that USP5-Beclin 1 axis is involved in lung tumorigenesis.

The main novelty of this study, in my opinion, is on USP5, and how USP5 regulates Beclin 1 in the context of lung cancer and oncogenic Ras mutations. The Beclin 1-p53 connections in senescence and cancer have already been reported.

A major strength of this manuscript is the rigor of the presented results. The observations are robust and supported by multiple models. With few exceptions, the experiments were designed in a meticulous manner, incorporating proper controls and statistical analyses throughout the study.

This reviewer has the following suggestions:

1. The ubiquitination experiments presented in Fig. 2 g, h, i should be performed under denature IP condition. This is because native IP can bring down ubiquitin signals from interacting proteins and may not represent the ubiquitin from the IP'ed target.
2. While the authors showed co-IP results suggesting a Beclin 1-p53-MDM2 complex, the direct binding partner for Beclin 1 is unclear to this reviewer. These experiments can be strengthened by using in vitro translated proteins or purified protein fragments expressed from bacteria. I understand that these interactions could be facilitated by modifications in the cells. Either way, the authors are encouraged to clarify this.
3. Junying Yuan group reported that Beclin 1 deficiency leads to reduced p53 protein level (PMID: 21962518). The present study showed that disrupting Beclin 1 stabilizes p53 in lung cancer. How to reconcile these contradictory results?

Minor:

1. There are several studies on USP5 in lung cancer (PMID: 34741014, PMID: 34858787, PMID: 32477134 and PMID: 30555744). The authors may consider mentioning some of these studies.
2. Typo in line 122 and line 149.

This review was submitted by Zhixun Dou.

Rebuttal

Reviewer #1

Remarks to the Author:

This is an excellent and thoughtful manuscript that characterizes the relationship between autophagy and senescence in lung cancer development. The work further identifies interaction(s) between USP5, Beclin and p53 in the regulation of autophagy and senescence and raises the possibility that USP5 influences clinical outcomes in lung cancer patients. The studies presented utilize appropriate technical approaches, particularly cells with expression of recombinant/mutant proteins and shRNA silencing to interrogate the putative relationships and impact on senescence and clonogenic survival. These latter data are particularly robust and convincing of the tumor growth repressive impact of senescence.

Response: We are grateful to the reviewer for the helpful comments and suggestions. We have addressed all of the issues raised by the reviewer.

Reviewer 1 Specific Comments #1:

A few deficiencies that likely could be readily corrected are indicated below.

"The Western blots should be accompanied by quantification of the protein bands. This is a problem, particularly in the case of LC3I to II conversion, where this indication of autophagy does not always appear to reflect the modulation of autophagy that the authors propose."

Response: We sincerely appreciate the reviewer's suggestion. As suggested, we performed densitometry of the LC3-II protein bands and then normalized them to corresponding GAPDH protein bands as it was often used in the field (Nature, 2017, PMID: 28445460; Nat Med, 2016, PMID: 27841876). The graphs showed quantitative data derived from three independent western blot experiments (revised Figure 1b, 1c, 1f, 4j, 6a, 6d and revised Supplementary Figure S2d, S2g, S5j). In addition, we also examined the p62/SQSTM1 by western blot analyses or IHC (revised Figure 1b, 1c, 1f, 4j, 6a, 6d and revised Supplementary Figure S2d, S2g, S5j) and LC3 puncta by IF to reflect the modulation of autophagy (revised Figure 1a). The conclusions from the quantifications support our original conclusions.

Reviewer 1 Specific Comments #2:

"A critical finding is that KRAS/G12D up regulates USP5 to promote autophagy, which prevents the cells from entering into senescence, and facilitates transformation. However, it appears that only a single time point is presented in support of this conclusion in Figure 5. This fundamental observation requires substantial confirmation."

Response: To address this issue, we established stable H292 cells harboring doxycycline-inducible KRas^{G12V}. We observed that induction of KRas^{G12V} expression was readily detected after 48 hours upon doxycycline, concomitant with increased expression of USP5, Beclin 1, and LC3-II, as well as down-regulation of p53 and p62/SQSTM1 (revised Figure 5a), in keeping with our previous observations. Notably, the increased enzymatic activities of USP5 were clearly observed at 12 hours after KRas induction, concomitant with increased cellular ROS levels (revised Supplementary Figure S5a). Thus, the induction of KRas^{G12V} expression leads to the elevation of ROS, and an increase of USP5 deubiquitinase activity, resulting in the modulation of downstream protein expression including Beclin 1, p53, p62/SQSTM1, and LC3-II, suggesting that KRas modulates USP5 deubiquitinase activity (revised manuscript, lines 290-296).

Reviewer 1 Specific Comments #3:

"The authors refer to decreased p62 on line 221, but there is no reference to where this information could be found in a figure. It is further unclear why p62 levels are not assessed routinely, for instance in Figure 4j(find p62 WB). It should be noted that p62 is routinely referred to as SQSTM1."

Response: We apologize for the unclear description and for the missed examination of p62 protein levels in the relevant experiments. The original statement "*decreased p62 on line 221*" could be found in the original Supplementary Figure S3b (revised Supplementary Figure S3c). Also, per suggestion, we used p62/SQSTM1 in the revised manuscript.

Regarding to the missing p62 protein levels, we performed a new set of western blotting using the same total cell lysates stored in -80 °C freezer. As shown in the revised Figures (Figure 1b, 1c, 1f, 6a, 6d and revised Supplementary Figure S2d, S2g, S5j), silencing of USP5 led to significantly increased p62/SQSTM1 protein expression, which could be effectively rescued by ectopic expression of Beclin 1, but not Beclin 1^{S90+93A} or Beclin 1^{L184+187A}, both of which are defective in promoting autophagy, indicating that silencing of USP5 inhibits autophagy in a cytoplasmic Beclin 1-dependent manner.

Reviewer 1 minor comment #1:

"Minor issue: Lines 40-41. The observations relating to the LC3 puncta could be explained in somewhat greater detail."

Response: We sincerely appreciate the reviewer's suggestion. Red fluorescent protein (RFP)-GFP tandem fluorescent-tagged LC3 (RFP-GFP-LC3) used in this work is considered as a reporter to monitor the autophagy flux. Since GFP-LC3, but not RFP-LC3, loses fluorescence due to lysosomal acidity and degradative conditions, the GFP and RFP signals derived from RFP-GFP-LC3 reflect lysosomes before fusion, while the RFP signal only reflects the lysosome fused with

autophagosome and increased autophagy (Autophagy, 2007, PMID: 17534139). In this revised manuscript, we quantitated the GFP-LC3 puncta (autophagosome) and RFP-LC3 puncta (autolysosomes) in A549-RFP-GFP-LC3 or H292-RFP-GFP-LC3 cells expressing USP5 or USP5^{W209A} mutation via QuPath (total > 200 cells per sample) (revised Figure 1a). Our data showed that ectopic expression of wild-type USP5, but not USP5^{W209A}, significantly increased both autophagosome formation (GFP-RFP-LC3 puncta) and autolysosome fusion (RFP-LC3 puncta). We hope that these descriptions in the revised version are clearer than before (revised manuscript, lines 140-146).

Reviewer 1 minor comment #2:

"The Discussion section is somewhat rambling and difficult to follow and could be tightened up, perhaps by having sections with subheadings. The authors could also consider addressing previous literature where the relationship between autophagy and senescence did not follow the same pattern as in the current work; that is, autophagy may have been reported to promote senescence or be required for senescence, rather than preventing senescence."

Response: We appreciate the reviewer's suggestions. We have reorganized the Discussion. We also added more literature discussion on the relationship between autophagy and senescence (revised manuscript, lines 455-461).

Reviewer #2

Remarks to the Author:

In this study, the authors investigate the role of USP5-Beclin1 axis in KRAS driven tumor progression. This topic is novel and interesting and the methodology is sound. The authors found that knockdown of USP5 or Beclin1 increase senescence and suppressed tumor growth in mice. Mechanistically, USP5 is stabilized by dimerization that can be induced by ROS. Stabilized USP5 improves Beclin1 protein stability, leading to degradation of p53. Knockdown of USP5 or Beclin1 promotes p53-dependent senescence and inhibits autophagy, resulting in suppression of KRAS driven NSCLC growth. This study opens a new window into developing strategies to target USP5 for cancer therapy. There are, however, some weak points in the paper that needs to be clarified to obtain sufficient support for some of the conclusions.

Response: We are grateful to the reviewer for the comments. We have worked hard to follow the advice of the reviewer and now provide new data and revisions in order to address all of the points raised by this reviewer.

Reviewer 2 comment #1:

"In Fig1e, it seems that knockdown of USP5 by two shRNAs is toxic for cell colony growth, however, the selectively specific inhibitor of USP5, WP1130 shows similar effect on Beclin1, p53 and p21 as shUSP5, but it does not result in similar suppression of cell growth. Are there off target effects of the shUSP5s or is the inhibitor ineffective in inhibiting USP5?"

Response: WP1130 (Degrasyn) is a selective deubiquitinase inhibitor for several DUBs, including USP5, USP9x, USP14, UCH-L1, and UCH37 (Cancer Research, 2010, PMID: 21045142). The reason that WP1130 did not result in similar suppression of cell growth compared to shUSP5 was most likely due to the lower dose of WP1130 used in our colony formation assays. In our new experiments, we treated A549 or H292 cells with an increasing dose of WP1130 (0, 2, 4, 6, 8, 10 μ M) in the colony formation assays, similar to a study (Oncotarget, 2017, PMID: 28446729). As shown in the revised Supplementary Figure S2f, treatment with 8 μ M WP1130 exhibited a dramatic suppression of cell growth similar to the inhibitory effects of knockdown of *USP5*. To exclude the off-target effects of the shUSP5s, we designed four different shRNAs targeting different regions of *USP5* mRNA. All of four shUSP5 showed a similar effect on the expression of Beclin1, p53 and p21 (revised Supplementary Figure S2a).

Reviewer 2 comment #2:

"The authors highlight the role of USP5-Beclin1 axis for overriding p53-dependent senescence in KRAS-driven tumors. However, the cell models used here are A549 (KRAS-G12S) and H292 (KRAS WT). The results of

manipulating USP-Beclin1 axis in both cell lines are the same. Does this mean that the USP-Beclin1 axis functions in lung cancer cell regardless of KRAS mutation? If so, please explain how."

Response: Our results indicate that activation of the USP5-Beclin1 axis can function to override p53-dependent senescence in lung cancer cells regardless of the *KRAS* status. In this study, we showed that an activating mutation of *KRAS* elevates intracellular ROS, leading to ROS-dependent oxidization and dimerization of USP5 protein through C195 disulfide formation, resulting in stabilization and activation of USP5 (revised Figure 5). Therefore, USP5 is a critical downstream effector of activated *KRAS*. Although we have not explored other means of activation of USP5, it is plausible that activation of USP5 via ROS can lead to modulation of Beclin 1-p53 axis in the absence of *KRAS* mutation (revised manuscript lines 300-310).

Reviewer 2 comment #3:

"Similarly, it would be important to define the role of the axis in p53 WT and p53-mutant/ko tumor cells, regardless of KRAS mutation."

Response: We agree that it is an important issue to investigate the USP5-Beclin 1-p53-senescence axis in p53 WT, p53 mutant or p53 null tumor cells. We therefore examined the effects of ablation of USP5 or Beclin 1 on the SA- β -gal staining in p53-mutant/null lung cancer cells, including SK-LU-1 (*KRAS*^{G12D};*p53*^{H193R}), H358 (*KRAS*^{G12C};*p53*^{-/-}), H1299 (*NRAS*^{Q61K};*p53*^{-/-}) and H1975 (*KRAS*^{WT};*p53*^{R273H}). Silencing of either *USP5* or *Becn1*, again, led to senescence as reflected by SA- β -gal staining in A549 cells (*KRAS*^{G12S};*p53*^{WT}) or H292 cells (*KRAS*^{WT};*p53*^{WT}). In contrast, silencing of either *USP5* or *Becn1* was unable to induce cellular senescence in any cancer cells harboring a mutant *p53* allele or *p53* null (revised Supplementary Figure S3b and S4a), which is consistent with the important role of p53 in mediating senescence (Oncogene, 2013, PMID: 23416979). These results indicate that wild-type p53 is indispensable for cellular senescence induced by inhibition of USP5 or Beclin 1 (revised manuscript lines 221-222; lines 240-242).

Reviewer 2 comment #4:

"The authors show nicely in vivo ubiquitination assay results in Fig3. Please explain whether this assay was performed under denaturing conditions. From the Methods in the manuscript, it seems the assay was performed with a normal Co-IP protocol, and not under denaturing conditions. As the results from ubiquitination assay are key parts of mechanism work, it will be important to explain and motivate in detail that how the in vivo ubiquitination assay was performed."

Response: We apologize for the unclear description. The in vivo ubiquitination assays performed in the original Fig3 were not under denaturing conditions. As the reviewer suggested, we re-performed all of *in vivo* ubiquitination assays

under denaturing conditions and we reached the same conclusions (revised Figure 2g, 2h, 2i and Supplementary Figure S4i, S4j, S5i).

Reviewer 2 comment #5:

"Then, an in vitro ubiquitination assay related to the Fig 3 should be performed."

Response: We have repeated the *in vitro* ubiquitination assay under denaturing conditions (revised Figure 5l).

Reviewer 2 comment #6:

"The Co-IP bands in Fig4m are distinctly and clearly displayed. The bands for p53 and IgG should be very similar in size. Thus, how did you technically go about separating them?"

Response: To specifically detect p53 protein while avoiding the interference of IgG, we adopted an IP-Western procedure used in the literature (Nature Commun, 2021, PMID: 34635651; Nature Cell Biology, 2015, PMID: 26280536). Briefly, we used a mouse p53 antibody (Santa Cruz, sc-126) to immunoprecipitate the p53 protein, and then we use a primary rabbit antibody specific for p53 (Santa Cruz, sc-6243) and HRP-conjugated goat against rabbit polyclonal antibody as the secondary antibody in the western blotting.

Reviewer 2 comment #7:

"In Fig5a and c, overexpression of KRAS-G12V upregulated the expression of USP5. By comparing the ratio between HA-Ub-Vs-USP5 and USP5, it seems that the increased USP5 activity was caused by increased protein expression of USP5. Hence, it seems a stretch to conclude that "KRAS-G12V significantly increased USP5 deubiquitinating enzymatic activity". In Fig5b, ROS inducers, H2O2, and PL clearly increased USP5 activity by comparing the ratio between HA-Ub-Vs-USP5 and USP5. The addition of NAC can't change the USP5 ratio. The authors should explain this further. It would also be informative if the authors include an experiment where cells that do not overexpress KRAS-G12V are incubated with NAC."

Response: This is an important issue whether *KRAS*^{G12V} can elevate the USP5 deubiquitinating enzymatic activity without accumulation of USP5 protein. To address this issue, we established stable H292 cells harboring doxycycline-inducible *KRas*^{G12V}, which were subjected to induce *KRas*^{G12V} expression upon doxycycline. The results showed that the deubiquitinating enzymatic activity of USP5 was elevated 12 hours after *KRas* induction, concomitant with increased cellular ROS levels, while the protein expression of Ras, USP5 and Beclin 1 were evidently elevated at 2 days after induction, together with reduced expression of p53 and p62/SQSTM1 in a

time-dependent manner (revised Figure 5a). These results indicate that activation of KRas^{G12V} leads to the elevation of ROS, resulting in increased deubiquitinating enzymatic activity of USP5 followed by the accumulation of USP5 protein (revised manuscript lines 290-297).

As the reviewer suggested, we treated H292 cells (*KRAS*^{WT}) with NAC. The results showed that treatment of NAC significantly decreased both basal levels of USP5 protein and activity (revised Figure S5c; revised manuscript lines 307-310).

Reviewer 2 comment #8:

"In Fig5l, shortly exposed bands in USP5-HA+KRAS-G12V sample should be added. If possible, significance levels should be analyzed in quantifications of CHX treatment results in Fig1b and c, Fig3a, Fig5d and l, and FigS4d."

Response: We appreciate the reviewer's suggestion. Accordingly, we added the shortly exposed bands (revised Figure 5m). Significance levels were analyzed in quantifications of CHX treatment results (revised Figure 2b, 2c, 3a, 5d, 5m, and Supplementary Figure S4e).

Reviewer 2 comment #9:

"The authors propose that USP5 and Beclin1 can regulate p53 stability and promote MDM2-mediated p53 degradation. A ubiquitination assay for p53 should be performed to directly confirm MDM2-mediated proteasomal-dependent degradation of p53, for example, by depleting USP5 or Beclin1."

Response: We sincerely appreciate the reviewer's suggestion. Accordingly, we examined the ubiquitination levels of p53 in either USP5- or Beclin 1-ablated 293FT cells transiently expressed p53-Myc and MDM2-HA. Ablation of *USP5* or *BECN1* significantly decreased the ubiquitin levels of p53, which could be completely rescued by ectopic expression of MDM2 (revised Supplementary Figures S4i and S4j).

Reviewer 2 comment #10:

"It is known that inhibition of USP5 induces apoptosis. Considering the role of p53 in cell apoptosis, how do authors exclude the possibility that the suppression of tumor growth by depletion of USP5 was not caused by increased apoptosis?"

Response: To address this issue, we examined apoptosis in USP5-depleted A549 or H292 cells by western blotting and FACS analyses. Our results showed that the depletion of USP5 had little effect on the expression of apoptotic biomarkers (cleaved PARP or Caspase-3) (revised Supplementary Figure S2b). In addition, FACS analyses showed that depletion of USP5 barely caused apoptosis in A549 and H292 cells (revised Supplementary Figure S2c; revised manuscript lines 276-278).

Reviewer #3

Remarks to the Author:

The manuscript by Juan Li et al discovered that USP5-Beclin 1 axis is critically involved in lung cancer cell senescence and tumorigenesis. Using a series of elegant experiments including cell biology, mechanistic studies, animal models and patient samples, the authors reported that USP5 stabilizes Beclin 1, which regulates p53 stability and senescence. Further, they showed that oncogenic Ras modulates USP5 via ROS, and that USP5-Beclin 1 axis is involved in lung tumorigenesis.

The main novelty of this study, in my opinion, is on USP5, and how USP5 regulates Beclin 1 in the context of lung cancer and oncogenic Ras mutations. The Beclin 1-p53 connections in senescence and cancer have already been reported.

A major strength of this manuscript is the rigor of the presented results. The observations are robust and supported by multiple models. With few exceptions, the experiments were designed in a meticulous manner, incorporating proper controls and statistical analyses throughout the study.

Response: We are grateful to the reviewer for the comments.

Reviewer 3 Comments #1:

"The ubiquitination experiments presented in Fig. 2 g, h, i should be performed under denature IP condition. This is because native IP can bring down ubiquitin signals from interacting proteins and may not represent the ubiquitin from the IP'ed target."

Response: We sincerely appreciate the reviewer's suggestion. As suggested, we have performed all of the ubiquitination experiments under denaturing IP conditions (revised Figure 2g, 2h, 2i and Supplementary Figure S4i, S4j, S5i).

Reviewer 3 Comments #2:

"While the authors showed co-IP results suggesting a Beclin 1-p53-MDM2 complex, the direct binding partner for Beclin 1 is unclear to this reviewer. These experiments can be strengthened by using in vitro translated proteins or purified protein fragments expressed from bacteria. I understand that these interactions could be facilitated by modifications in the cells. Either way, the authors are encouraged to clarify this."

Response: We appreciate the reviewer's constructive suggestion. Accordingly, we performed a series of new experiments using purified recombinant His-Flag-p53,

His-Beclin 1 and MDM2 from *E. coli*. The in vitro binding assay showed that Beclin 1 could directly bind to p53 and MDM2, respectively (revised Figure 4n-4o). Furthermore, Beclin 1 could facilitate the interaction between p53 and MDM2 in a dose-dependent manner in vitro (revised Figure 4q).

Reviewer 3 Comments #3:

"Junying Yuan group reported that Beclin 1 deficiency leads to reduced p53 protein level (PMID: 21962518). The present study showed that disrupting Beclin 1 stabilizes p53 in lung cancer. How to reconcile these contradictory results?"

Response: We thank this reviewer for asking this question. We think that the different mechanisms may be due to the complex role of Beclin 1 in cancer development in a cancer- and cell-type-specific manner. Our study focuses on the cell types that can be induced to undergo senescence while most cancer lines cannot be induced to undergo senescence (therefore, they are already defective in senescence mediated by tumor suppression mechanism). Thus, the mechanisms revealed by this study may be relevant only to the cell types that can undergo senescence.

The notion that Beclin 1 can function as a tumor suppressor protein is supported by several well-known studies, including the observations that endogenous Beclin 1 protein expression is frequently low in human breast epithelial carcinoma cell lines and tissue, but is expressed ubiquitously at high levels in normal breast epithelia (Nature, 1999, PMID: 10604474); *BECN1* is monoallelically lost in 40% to 75% of human breast, and ovarian cancers (Nature, 1999, PMID: 10604474). The *Becn1*^{+/-} mice are prone to the development of liver and lung tumors and lymphomas (J Clin Invest, 2003; PMID: 14638851; PNAS, 2003; PMID: 14657337).

However, large-scale genomic analyses of human cancers have failed to identify recurrent mutations in *BECN1* (Nature, 2014, PMID: 24390350; Science, 2013, PMID: 23539594), implying that *BECN1* may not be a classical tumor suppressor in most human cancers (Mol Cancer Res, 2014, PMID: 24478461). Additionally, *BECN1* allelic loss is confounded by its location adjacent to *BRCA1* on human chromosome 17q21, raising the possibility that loss of *BECN1* in human cancers may be associated with the loss of *BRCA1* in human breast and ovarian cancers (Mol Cancer Res, 2014, PMID: 24478461). Notably, in *PALB2* loss mouse models for hereditary breast cancer, allelic loss of *Becn1* promotes p53 activation and reduces tumorigenesis (Cancer Discov, 2013, PMID: 23650262), suggesting an intimate relation of p53 in Beclin 1-mediated regulation of tumorigenesis.

In addition, abundant evidence indicates that autophagy plays an important role in activated Ras-induced tumorigenesis (J Clin Invest, 2015, PMID: 25654549), in which p53 is intimately involved. The *Kras*^{G12V}-driven salivary duct carcinoma progression is severely hindered by the deficiency of the autophagy-essential gene *Atg5*

(Autophagy, 2018, PMID: 29956571). Similarly, *Kras*^{G12D}-driven NSCLC is inhibited by *Atg7* deletion. *Atg7*-deficient tumors show prematurely induced p53 and proliferative arrest (Genes Dev, 2013, PMID: 23824538). At later stages of tumorigenesis, *Atg7* deficiency causes p53 activation, accumulation of defective mitochondria, proliferative defects, and reduced tumor burden (Cancer Discov, 2013, PMID: 23965987).

Taken together, our study revealed the USP5-Beclin 1 axis in regulating p53-dependent senescence, which has not been addressed by previous studies (revised manuscript 446-452).

Reviewer 3 minor comment #1

"There are several studies on USP5 in lung cancer (PMID: 34741014, PMID: 34858787, PMID: 32477134 and PMID: 30555744). The authors may consider mentioning some of these studies."

Response: We thank the reviewer's suggestion. Accordingly, we have cited these references and integrated in the revised manuscript (revised manuscript lines 81-83).

Reviewer 3 minor comment #2

"Typo in line 122 and line 149."

Response: We apologize for typos, which have been corrected (line 122 and line 155). We have carefully proofread the revised manuscript to minimize spelling and grammar errors.

REVIEWERS' COMMENTS

Reviewer #1 (Remarks to the Author):

The authors have addressed all of the concerns raised in the previous review thoroughly and conscientiously, providing additional data in support of the conclusions relating to USP5, Beclin, p53, autophagy and senescence.

Reviewer #2 (Remarks to the Author):

The authors have appropriately addressed the questions related to the first comments. However, there are a few issues remaining that should be addressed.

In Fig.5a, the authors state that mutant KRAS expression was induced by doxycycline, leading to increased ROS production and USP5 deubiquitinating enzymatic activity after 12 hours, but RAS protein levels are not detected by WB. All the protein expression changes shown in Fig.5a followed the RAS expression after 2 days except 'HA-UB-Vs-USP5'. The authors should exclude the potential effect of Doxycycline itself on 'HA-UB-Vs-USP5'. And explain the functional relationship between HA-UB-Vs-USP5 upregulation at 12 hours and the induction of mutant (but un-detectable) KRAS.

A second issue to address is that in Fig.5b ROS inducers H₂O₂ and PL clearly increased USP5 activity by comparing the ratio between HA-UB-VS-USP5 and USP5, but in Fig.5c, induction of mutant KRAS did not change the ratio between HA-UB-VS-USP5 and USP5; thus, increased HA-UB-VS-USP5 seems to be induced by increased USP5 expression. Treatment of NAC on H292 cells in FigS5c clearly reduced USP5 expression, and it seems possible that the reduced USP5 activity could simply be caused by reduced USP5 protein expression. Does mutant KRAS expression and ROS inducers upregulate USP5 activity via the same mechanism? This issue should be addressed in the discussion.

Reviewer #3 (Remarks to the Author):

The authors have satisfactorily addressed my critiques. I support publication of this manuscript at Nature Communications.

Reviewer #2 (Remarks to the Author):

The authors have appropriately addressed the questions related to the first comments. However, there are a few issues remaining that should be addressed.

In Fig.5a, the authors state that mutant KRAS expression was induced by doxycycline, leading to increased ROS production and USP5 deubiquitinating enzymatic activity after 12 hours, but RAS protein levels are not detected by WB. All the protein expression changes shown in Fig.5a followed the RAS expression after 2 days except 'HA-UB-Vs-USP5'. The authors should exclude the potential effect of Doxycycline itself on 'HA-UB-Vs-USP5'. And explain the functional relationship between HA-UB-Vs-USP5 upregulation at 12 hours and the induction of mutant (but un-detectable) KRAS.

A second issue to address is that in Fig.5b ROS inducers H₂O₂ and PL clearly increased USP5 activity by comparing the ratio between HA-UB-VS-USP5 and USP5, but in Fig.5c, induction of mutant KRAS did not change the ratio between HA-UB-VS-USP5 and USP5; thus, increased HA-UB-VS-USP5 seems to be induced by increased USP5 expression. Treatment of NAC on H292 cells in FigS5c clearly reduced USP5 expression, and it seems possible that the reduced USP5 activity could simply be caused by reduced USP5 protein expression. Does mutant KRAS expression and ROS inducers upregulate USP5 activity via the same mechanism? This issue should be addressed in the discussion.

Response:

We thank the reviewer for raising this issue. In our experiments, we reproducibly showed that the USP5 enzymatic activity was activated at 12 h after doxycycline induction, as evidenced by the elevated HA-Ub-Vs-USP5, while the elevation of KRAS protein expression was more prominent at 2 days upon doxycycline induction. Notably, in previous reports using the doxycycline-inducible system, the doxycycline-induced protein expression can be detected by Western blotting onward from 12 h and steadily accumulated for more than 60 h (J Mol Endocrinol, 2016, PMID: 27099398). In particular, HRAS protein expression is induced on day 1 after doxycycline induction (Molecule Cell, 2011, PMID: 21353614). Thus, we think it is plausible that doxycycline-induced KRAS protein expression at the early stage (undetectable by western blotting at 12 hours) could elevate the intracellular ROS levels, which is indeed the case as shown by DCFA assays (Fig S5a).

Regarding the potential effect of Doxycycline itself on 'HA-UB-Vs-USP5', we searched the literature and could not find any reports showing the side effects of doxycycline at low doses. Notably, a study reported that doxycycline (at 15 µg/ml) can lead to increased p53 protein expression and cell cycle G1 arrest (Anticancer Research, 2009, PMID: 19846942). In our study, we used doxycycline (at 1 µg/ml) and observed no obvious abnormalities associated with doxycycline alone, including cell morphology, cell growth, or cell viability. Hence, we think that the effect of Doxycycline on 'HA-UB-Vs-USP5' is associated with doxycycline-induced KRAS expression and is highly

unlikely attributable to side effects.

In the revised manuscript, we better explained the functional relationship between HA-UB-Vs-USP5 upregulation at 12 hours and the induction of mutant (but un-detectable) KRAS in discussion (lines 490-493).

Regarding the molecular mechanisms by which activated KRAS and ROS inducers (H_2O_2 and piperlongumine) upregulate USP5 activity, our results strongly support the notion that elevated ROS is an underlying mechanism shared by KRAS and ROS inducers for the upregulation of USP5 enzymatic activity as well as USP5 protein expression (Fig 5a-5b, Fig S5a-S5c). With this regard, we appreciate the reviewer for making this issue clearer, i.e., that ROS promotes both USP5 activity and USP5 protein expression, the latter of which greatly contributes to the upregulation of USP5 activity. We have made this clearer with added discussion in the revised manuscript (lines 493-497).